# Discovery of long-range inhibitory signaling to ensure single axon formation

Tetsuya Takano[1], Mengya Wu[1], Shinichi Nakamuta[1], Honda Naoki[2], Naruki Ishizawa[1], Takashi Namba[1], Takashi Watanabe[3], Chundi Xu[1], Tomonari Hamaguchi[1], Yoshimitsu Yura[1], Mutsuki Amano[1], Klaus M. Hahn[3] & Kozo Kaibuchi[1]

A long-standing question in neurodevelopment is how neurons develop a single axon and multiple dendrites from common immature neurites. Long-range inhibitory signaling from the growing axon is hypothesized to prevent outgrowth of other immature neurites and to differentiate them into dendrites, but the existence and nature of this inhibitory signaling remains unknown. Here, we demonstrate that axonal growth triggered by neurotrophin-3 remotely inhibits neurite outgrowth through long-range $Ca^{2+}$ waves, which are delivered from the growing axon to the cell body. These $Ca^{2+}$ waves increase RhoA activity in the cell body through calcium/calmodulin-dependent protein kinase I. Optogenetic control of Rho-kinase combined with computational modeling reveals that active Rho-kinase diffuses to growing other immature neurites and inhibits their outgrowth. Mechanistically, calmodulin-dependent protein kinase I phosphorylates a RhoA-specific GEF, GEF-H1, whose phosphorylation enhances its GEF activity. Thus, our results reveal that long-range inhibitory signaling mediated by $Ca^{2+}$ wave is responsible for neuronal polarization.

[1] Department of Cell Pharmacology, Nagoya University Graduate School of Medicine, Nagoya 466-8550, Japan. [2] Imaging Platform for Spatio-Temporal Information, Graduate School of Medicine, Kyoto University, Kyoto 606-8315, Japan. [3] Department of Pharmacology, University of North Carolina, Chapel Hill, North Carolina 27599, USA. Correspondence and requests for materials should be addressed to K.K. (email: kaibuchi@med.nagoya-u.ac.jp)

Neurons are highly polarized cells that have two structurally and functionally distinct compartments: axons and dendrites[1–5]. Specific patterns of axonal elongation and dendritic formation are crucial for neuronal development and functions[6–8]. The processes responsible for neuronal polarization have been extensively studied using hippocampal neurons as a model system[4, 9]. Hippocampal neurons first extend several filopodia all around the cell body (stage 1). These neurons then generate multiple, morphologically similar immature neurites (i.e., minor neurites). These minor neurites repeatedly extend and retract (stage 2). A fast-growing neurite becomes an axon (stage 3), while the remaining minor neurites continue to undergo growth and retraction, thereby developing into dendrites at later stages (stage 4; day 4–7). Because axonal fate is stochastically determined in the absence of additional extracellular factors, this process is called "the stochastic model" of neuronal polarization.

Extracellular factors such as neurotrophins and insulin-like growth factor-1 play a critical role in neuronal polarization[10–13]. Among these factors, neurotrophins such as brain-derived

**Fig. 1** Local application of neurotrophins to an axon terminal induced remote minor neurite retraction. **a** Local application of neurotrophins induced remote minor neurite retraction. PBS, NT-3, or BDNF was locally applied to the axon terminal of a stage 3 hippocampal neuron, and the images following this application are presented. *Yellow* and *white arrowheads* indicate the retracting and resting minor neurites, respectively. Scale bar, 50 μm. **b–d** Time course of changes in the lengths of the axon (*blue*) and minor neurites (*red*) from a single neuron. **e, f** Axonal outgrowth (PBS = 14, NT-3 = 15, BDNF = 16 neurons from three independent experiments) and minor neurite outgrowth (PBS = 45, NT-3 = 39, BDNF = 31 neurites from three independent experiments) were measured. Error bars represent SEM. *$P < 0.05$ and **$P < 0.01$

neurotrophic factor (BDNF) and neurotrophin-3 (NT-3) act in autocrine or paracrine manners to regulate neuronal polarization in hippocampal neurons and the developing neocortex[10–12]. Neurotrophins are produced from neurons and amplified in one minor neurite, thereby leading to axon specification[10–12].

Neurotrophin receptors, designated Trks, are also selectively localized at the distal part of the axon and have been implicated in neuronal polarization[10, 12]. Attenuation of neurotrophins and/or Trks impairs neuronal polarization[11, 12], indicating that neurotrophin/Trk signaling is essential for neuronal polarization

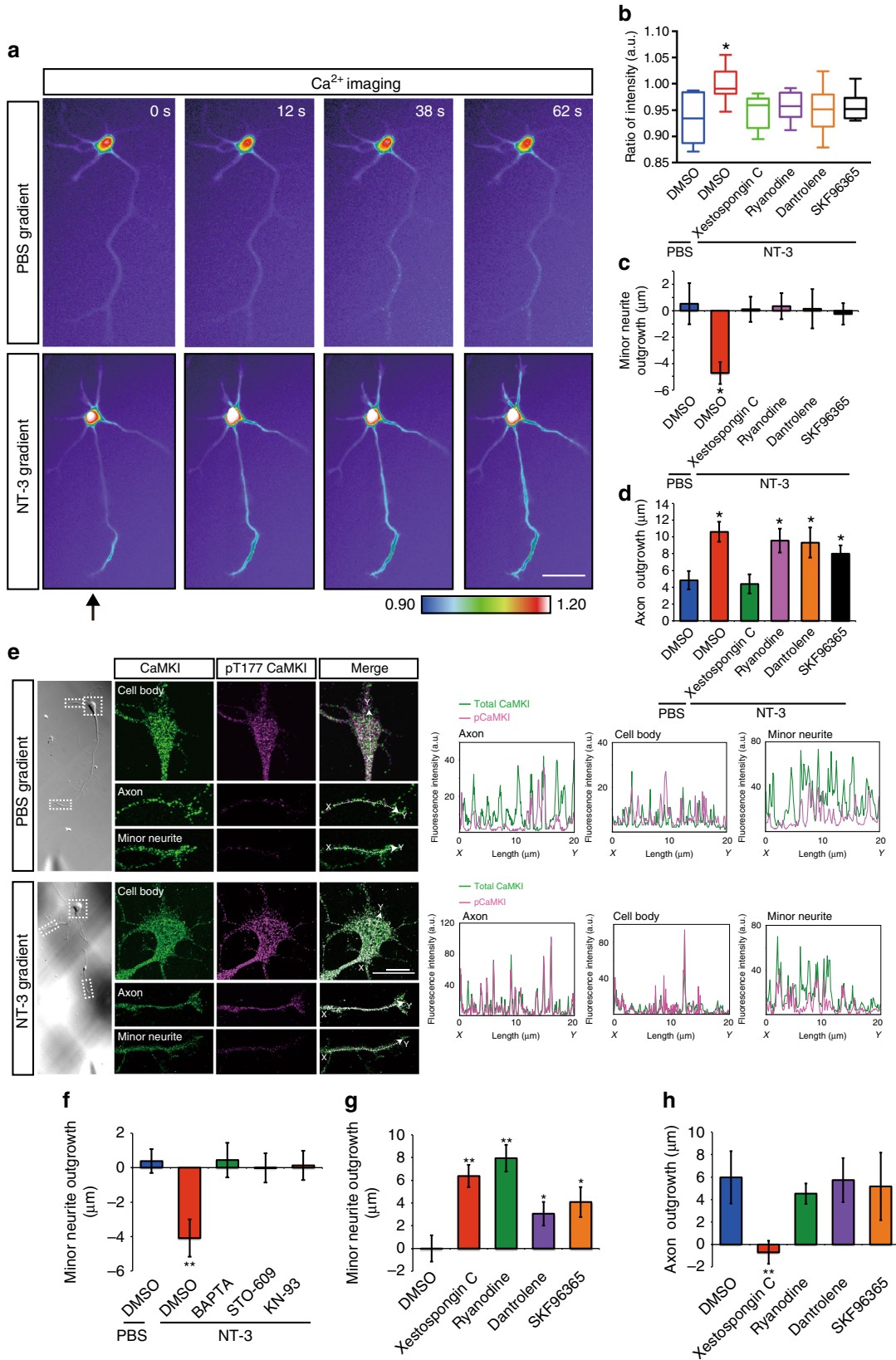

even in the stochastic model[14–17]. However, previous studies cannot explain how neurons generate only one axon and prevent multiple axons from forming.

We previously hypothesized that polarized neurons use a robust long-range inhibitory mechanism to generate only one axon, thereby determining dendritic specification[1, 3]. Once axonal fate is determined, the growing nascent axon of a stage 3 neuron is speculated to send a stronger long-range inhibitory signal to the other minor neurites, preventing the formation of unnecessary axons. However, the existence of long-range inhibitory signaling has not been explored and determined. Here, we discover a spatiotemporal long-range inhibitory signaling process that is mediated by unique $Ca^{2+}$ waves and guarantees proper neuronal polarization.

## Results

**Neurotrophinson an axon cause minor neurite retraction**. We previously reported that NT-3 derived from neurons are required for axon specification in stage 2 cultured hippocampal neurons[12]. We here examined whether the local amplification of NT-3 in a nascent axon enhances axon formation in polarized neurons (stage 3). We found that local application of a neutralizing antibody against NT-3 to the axon terminal inhibited axon outgrowth (Supplementary Fig. 1), indicating that the local amplification of NT-3 is required for axon formation in the stochastic model. Although long-range inhibitory signals may be produced from growing axons instage 3 cultured hippocampal neurons, we assumed that visualizing these signals in a static state poses practical difficulties. We hypothesized that local stimulation of nascent axons by neurotrophins would enhance the inhibitory signals and improve their visualization. To test this hypothesis, we locally exposed the axon terminals to gradients of NT-3 or BDNF for 45 min (Fig. 1a, left panel). Local application of NT-3 or BDNF to axon terminals remarkably increased axonal elongation compared with the control (phosphate-buffered saline [PBS])(Fig. 1b–e). Concurrently, the terminals of most of the minor neurites exhibited a backwards movement (Fig. 1a, yellow arrowheads), resulting in a shortening of minor neurite length (Fig. 1c, d, f). These results suggest that long-range inhibitory signaling triggered by neurotrophins exists and induces the retraction of minor neurites.

**Long-range $Ca^{2+}$ waves regulate minor neurite retraction**. The signals that propagate from the axon terminal to the cell body and/or the minor neurites that induce minor neurite retraction remain unknown. Because NT-3 is more abundant than BDNF in the developing neocortex[18], we focused on the functions of NT-3. We previously reported that NT-3 increases $Ca^{2+}$ concentrations at the growth cones of stage 2 hippocampal neurons[12]. We speculated that intracellular $Ca^{2+}$ mediates long-range inhibitory signaling. To investigate this possibility, hippocampal neurons were loaded with a $Ca^{2+}$ indicator (Cal-520 AM) and then subjected to real-time imaging. We observed that control neurons showed slight and transient increases in intracellular $Ca^{2+}$ in the axonal shaft (Fig. 2a and Supplementary Movie 1). Local application of NT-3 produced transient increase in $Ca^{2+}$ waves, which were propagated from the axon to the cell body and occasionally to the minor neurites (Fig. 2a and Supplementary Movie 2). To determine the source of these long-range $Ca^{2+}$ waves, we measured $Ca^{2+}$ concentrations in the cell bodies of hippocampal neurons treated with several specific inhibitors: The inhibitors (including inhibitors of ryanodine receptors, the most downstream source of cytoplasmic $Ca^{2+}$) abolished the NT-3-induced elevation of $Ca^{2+}$ concentration in the cell body (Fig. 2b). We next examined whether the long-range $Ca^{2+}$ waves generated from ryanodine receptors were imprecated in NT-3-induced minor neurite retraction. All the inhibitors abolished the minor neurite retraction induced by NT-3 (Fig. 2c). On the other hand, the NT-3-induced axonal elongation was mitigated by treatment with xestospongin C (IP₃ receptor inhibitor) but not by the other inhibitors (Fig. 2d), which was consistent with our previously reported study[12]. These results indicate that long-range $Ca^{2+}$ waves are generated from ryanodine receptors through $Ca^{2+}$-induced $Ca^{2+}$ release and are required for NT-3-induced minor neurite retraction.

Calmodulin-dependent protein kinase kinase (CaMKK) and its target calcium/calmodulin-dependent protein kinase I (CaMKI) have been implicated in neuronal polarization[12, 19, 20]. Therefore, to examine whether local stimulation of an axon with NT-3 increases CaMKI activity in the cell body of the same neuron, we immunostained locally stimulated neurons with anti-pT177-CaMKI, which represents the active form of CaMKI, and anti-CaMKI antibodies. The slight phospho-CaMKI was observed in the cell body of control neurons but not the axon and minor neurites (Fig. 2e). The local application of NT-3 to an axon markedly increased the phosphorylation of CaMKI in the cell body as well as the axon but not in the minor neurites (Fig. 2e). To examine whether the long-range $Ca^{2+}$ wave that led to the CaMKK/CaMKI pathway was responsible for NT-3-induced minor neurite retraction, we used the following specific inhibitors: BAPTA-AM ($Ca^{2+}$ chelator), STO-609 (CaMKK inhibitor) and KN-93 (CaMKI inhibitor). All of the inhibitors restored the minor neurite retraction induced by NT-3 to the levels observed in control neurons (Fig. 2f). We examined whether an increase in intracellular $Ca^{2+}$, which leads to $Ca^{2+}$-induced $Ca^{2+}$ release, induces minor neurite retraction using ionomycin ($Ca^{2+}$ ionophore)[21, 22]. Local application of ionomycin to the axon induced

**Fig. 2** NT-3 generated long-range $Ca^{2+}$ signaling from the axon to the cell body. **a** Long-range $Ca^{2+}$ wave. The relative change in the Cal-520 emission ratio (defined as R) was used as a measure of changes in $Ca^{2+}$ concentration. The pseudocolored images represent R after local application of PBS (*top*) or NT-3 (*bottom*) to the axon (*arrow*). Scale bars, 50 μm. **b** The mean amplitude of $R_{treat}/R_0$ for 90 s during local application of NT-3 in the presence of the indicated inhibitors (PBS = 10, NT-3 = 16, xestospongin C = 9, ryanodine = 15, dantrolene = 18, SKF96365 = 12 neurons from three independent experiments). **c**, **d** The axon was exposed to NT-3 in the presence of the indicated inhibitors, and then minor neurite outgrowth (PBS = 21, NT-3 = 21, xestospongin C = 26, ryanodine = 25, dantrolene = 25, SKF96365 = 24 neurites from three independent experiments) **c** and axonal outgrowth (PBS = 7, NT-3 = 9, xestospongin C = 9, ryanodine = 9, dantrolene = 8, SKF96365 = 8 neurons from three independent experiments) **d** were measured. **e** Local application of NT-3 increased the quantity of phospho-CaMKI in the cell body. After local application of PBS (*top*) or NT-3 (*bottom*), hippocampal neurons were immunostained with antibodies against CaMKI (*green*) and phospho-Thr177 of CaMKI (*magenta*). The merged images (*right panels*) are shown. The graph plots the fluorescence intensities of total CaMKI (*green*) and CaMKI phosphorylated at Thr177 (*magenta*) and in the line. Scale bars, 20 μm. **f** NT-3-induced minor neurite retraction was abolished by $Ca^{2+}$ signaling inhibitors. The axon was exposed to NT-3 in the presence of the indicated inhibitors, and minor neurite outgrowth was measured (PBS = 27, NT-3 = 31, BAPTA = 47, STO-609 = 45, KN-93 = 40 neurites from three independent experiments). **g**, **h** Local application of indicated inhibitors to the axon. Minor neurite outgrowth (DMSO = 42, xestospongin C = 32, ryanodine = 37, dantrolene = 35, SKF96365 = 32 neurons from three independent experiments) **g** and axonal outgrowth (DMSO = 14, xestospongin C = 11, ryanodine = 13, dantrolene = 13, SKF96365 = 12 neurons from three independent experiments) **h** were measured. Error bars represent SEM. *$P < 0.05$ and **$P < 0.01$

minor neurite retraction, whereas axonal elongation was not affected (Supplementary Fig. 2a–c). We also found that local application of ionomycin to the cell body induced minor neurite retraction, indicating that elevation of $Ca^{2+}$ concentration in the cell body is sufficient to retract minor neurites (Supplementary Fig. 2d, e). We next examined the effect of endogenous

long-range $Ca^{2+}$ propagations on minor neurite outgrowth and axon outgrowth using $Ca^{2+}$ signaling inhibitors. Local application of the ryanodine receptor inhibitors to axons induced minor neurite elongation (Fig. 2g). Notably, local application of xestospongin C but not the other inhibitors suppressed axonal outgrowth (Fig. 2h), indicating that the local $Ca^{2+}$ elevation

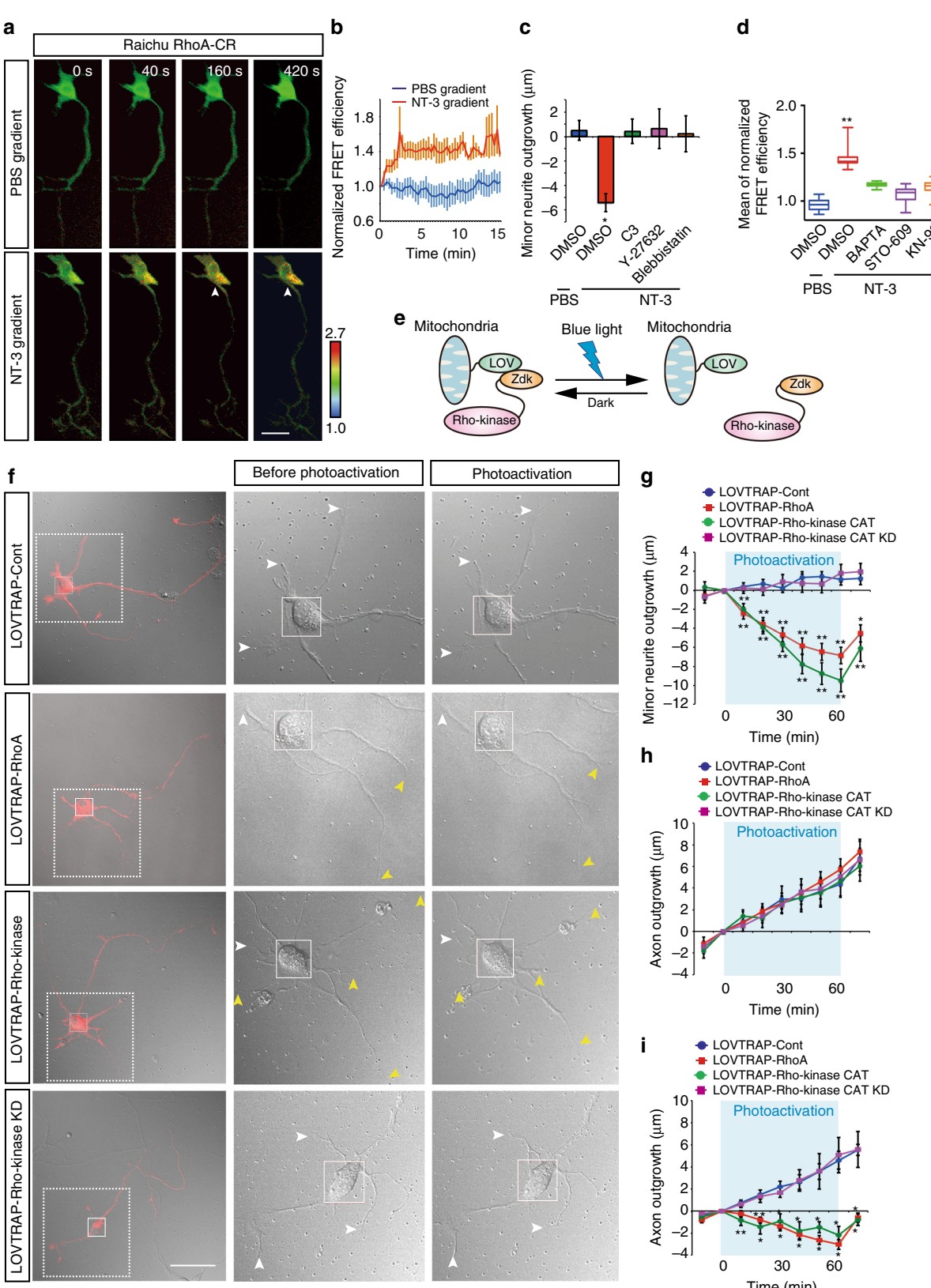

through IP$_3$ receptors is required for axon formation. We also found that local application of BAPTA-AM or KN-93 to the cell body induced minor neurite elongation (Supplementary Fig. 2d, e). Altogether, these results suggest that long-range Ca$^{2+}$ waves delivered from the axon terminal to the cell body are triggered by NT-3 and negatively regulate minor neurite outgrowth through the CaMKK/CaMKI pathway.

**Polarized activation of RhoA/Rho-kinase retracts neurites.** We next explored signaling downstream of the long-range Ca$^{2+}$ signaling. The Rho family GTPase RhoA has been shown to be a negative regulator of axon formation in stage 2 neurons, through its downstream effector Rho-kinase (ROCK)[23–27]. Therefore, we examined whether local stimulation of an axon with NT-3 could induce activation of RhoA using the intramolecular fluorescence-resonance energy transfer (FRET) probe Raichu-RhoA, which enables the monitoring of the balance between guanine nucleotide exchange factor (GEF) and GTPase-activating protein (GAP) activities and, therefore, serves as a surrogate marker for RhoA activation[28]. From the first time point (approximately 3 min) after local application of NT-3, RhoA activity markedly increased in the cell body (Fig. 3a, arrowheads and Supplementary Movie 3 and 4), and the increment of RhoA activity was subsequently sustained throughout the observation period (Fig. 3b). RhoA activation was also observed in the minor neurites of NT-3-stimulated neurons, but not in the axon terminal (Supplementary Fig. 3). To examine whether NT-3-induced RhoA activation was responsible for minor neurite retraction, we used the following specific inhibitors: C3 (RhoA inhibitor), Y-27632 (Rho-kinase inhibitor), and blebbistatin (myosin II inhibitor). All the inhibitors restored the minor neurite retraction induced by NT-3 to the levels observed in control neurons (Fig. 3c). Because activation of RhoA in the cell body (3–5 min) after NT-3 stimulation occurred later than the long-range Ca$^{2+}$ wave (~1 min) (Fig. 2a and Fig. 3a, b), we next determined whether NT-3-induced RhoA activation was mediated by long-range Ca$^{2+}$ signaling. We found that treatment with BAPTA-AM, STO-609 or KN-93 partially reduced NT-3-induced RhoA activation in the cell body (Fig. 3d), indicating that the polarized activation of RhoA/Rho-kinase is associated with long-range Ca$^{2+}$ signaling.

Optogenetic techniques are useful for analyzing the role of a molecule by activating it at particular times and/or subcellular locations through illumination[29]. We developed a photoactivatable Rho-kinase using the LOVTRAP optogenetic approach to enable local activation of the kinase[30]. In the dark, Rho-kinase CAT-Zdk1 was sequestered in mitochondria. During blue-light illumination, Rho-kinase Zdk1 was released from the mitochondria (Fig. 3e). Intermittent irradiation (5 s on/off cycle) for 30 min led to stress-fiber formation in COS-7 cells and membrane blebbing and cell contraction at the leading edge (line 2) in HeLa cells (Supplementary Fig. 4a–d and Supplementary Movie 5). In contrast, HeLa cells expressing a kinase-dead form of the Rho-kinase Zdk1 were not visibly affected by irradiation (Supplementary Fig. 4e, f). Altogether, these results validate the use of the LOVTRAP approach as a means to manipulate Rho-kinase activity in living cells.

Given that the minor neurite retraction induced by NT-3 (approximately 15 min) was preceded by activation of RhoA (3–5 min) (Figs 1a and 3a, b), we used the LOVTRAP-RhoA and LOVTRAP-Rho-kinase to evaluate whether local activation of RhoA or Rho-kinase in the cell body could induce minor neurite retraction. To accomplish this, we transfected LOVTRAP-RhoA, LOVTRAP-Rho-kinase, or the kinase-dead mutant into hippocampal neurons. The cell bodies of the neurons were illuminated at a wavelength of 488 nm. Neurons expressing mitochondrially anchored LOV2 with mCherry-Zdk1or with kinase-dead mCherry-Rho-kinase CAT-Zdk1 were not affected by blue-light illumination (Fig. 3f–h, and Supplementary Movie 6 and Supplementary Movie 7). Under these conditions, the photoactivation of LOVTRAP-RhoA or LOVTRAP-Rho-kinase in the cell body significantly induced rapid minor neurite retraction from the first time point (approximately 10 min) (Fig. 3f, yellow arrowheads, Fig. 3g and Supplementary Movie 8). Interestingly, LOVTRAP-RhoA or LOVTRAP-Rho-kinase did not affect axon length (Fig. 3h). We also investigated the effect of active RhoA/Rho-kinase on axonal outgrowth by photoactivation of RhoA or Rho-kinase in the distal part of the axon. We found that photoactivation of RhoA or Rho-kinase, but not the kinase-dead mutant of Rho-kinase, suppressed axonal elongation compared with control neurons (Fig. 3i), indicating that axonal outgrowth is associated with a small amount of active RhoA/Rho-kinase in the distal part of the axon. These results suggest that polarized activation of RhoA/Rho-kinase in the cell body induced by long-range Ca$^{2+}$ waves is sufficient to retract minor neurites but not to affect axon outgrowth.

**Rho-kinase diffuses in the neural shaft.** To test whether photoactivated Rho-kinase could diffuse from cell bodies into short minor neurites but not into long axons, we developed a mathematical model[31]. In the model, Rho-kinase undergoes photoactivation in the cell body, passively diffuses along a neurite, and is then inactivated by becoming trapped in mitochondria and

**Fig. 3** Polarized activation of RhoA/Rho-kinase in the cell body was required for minor neurite retraction. **a** Spatio-temporal activation of RhoA in neurons. PBS (*top*) or NT-3 (*bottom*) was locally applied to the axon terminal (*arrow*) of a neuron expressing Raichu-RhoA-CR. The pseudocolored images represent the Raichu-RhoA-CR emission ratio. *Arrowhead* indicates the activation of RhoA at the cell body. Scale bar, 20 μm. **b** Time course of changes in FRET efficiency in the cell body. **c** NT-3-induced minor neurite retraction was abolished by RhoA/Rho-kinase signaling inhibitors. The axon was exposed to NT-3 in the presence of the indicated inhibitors, and minor neurite outgrowth was measured (DMSO (PBS) = 16, DMSO (NT-3) = 22, C3 = 23,Y-27632 = 28, Blebbistatin = 22 neurites from three independent experiments). **d** The mean amplitude of increases in FRET efficiency for 2 min during local application were measured. The FRET efficiency in the cell body was analyzed after treatment with vehicle (control) or the indicated inhibitors in the presence of NT-3 (DMSO (PBS) = 37, DMSO (NT-3) = 37, BAPTA = 37, STO-609 = 37, KN-93 = 37 neurons from three independent experiments). **e** Cartoon representation of the photoactivatable Rho-kinase (LOVTRAP-Rho-kinase). **f** Photoactivation of RhoA or Rho-kinase in the cell body induced minor neurite retraction. The cell body of a neuron coexpressing NTOM20-LOV2 with mCherry-Zdk1 (LOVTRAP-Cont), mCherry-Zdk1-RhoA Q63L (LOVTRAP-RhoA), mCherry-Rho-kinase CAT-Zdk1 (LOVTRAP-Rho-kinase), or mCherry-Rho-kinase CAT KD-Zdk1 (LOVTRAP-Rho-kinase KD) is illuminated in the 20-μm square. Representative images of neurons by photoactivation for 60 min are shown. *Yellow* and *white arrowheads* indicate the retracting and resting minor neurites, respectively. **g**, **h** Time course of changes in the lengths of minor neurites **g** and the axon **h** from a single neuron expressing LOVTRAP-Cont, LOVTRAP-RhoA, LOVTRAP-Rho-kinase, or LOVTRAP-Rho-kinase KD (LOVTRAP-Cont = 40, LOVTRAP-RhoA = 23, LOVTRAP-Rho-kinase CAT = 44, LOVTRAP-Rho-kinase CAT KD = 36 neurites from five independent experiments). **i** Photoactivation of RhoA or Rho-kinase in the axon induced axonal retraction. The distal part of the axon of a neuron expressing LOVTRAP-Cont, LOVTRAP-RhoA, LOVTRAP-Rho-kinase or LOVTRAP-Rho-kinase KD is illuminated (LOVTRAP-Cont = 8, LOVTRAP-RhoA = 9,LOVTRAP-Rho-kinase CAT = 9, LOVTRAP-Rho-kinase CAT KD = 9 neurons from three independent experiments). Error bars represent SEM. *$P < 0.05$ and **$P < 0.01$

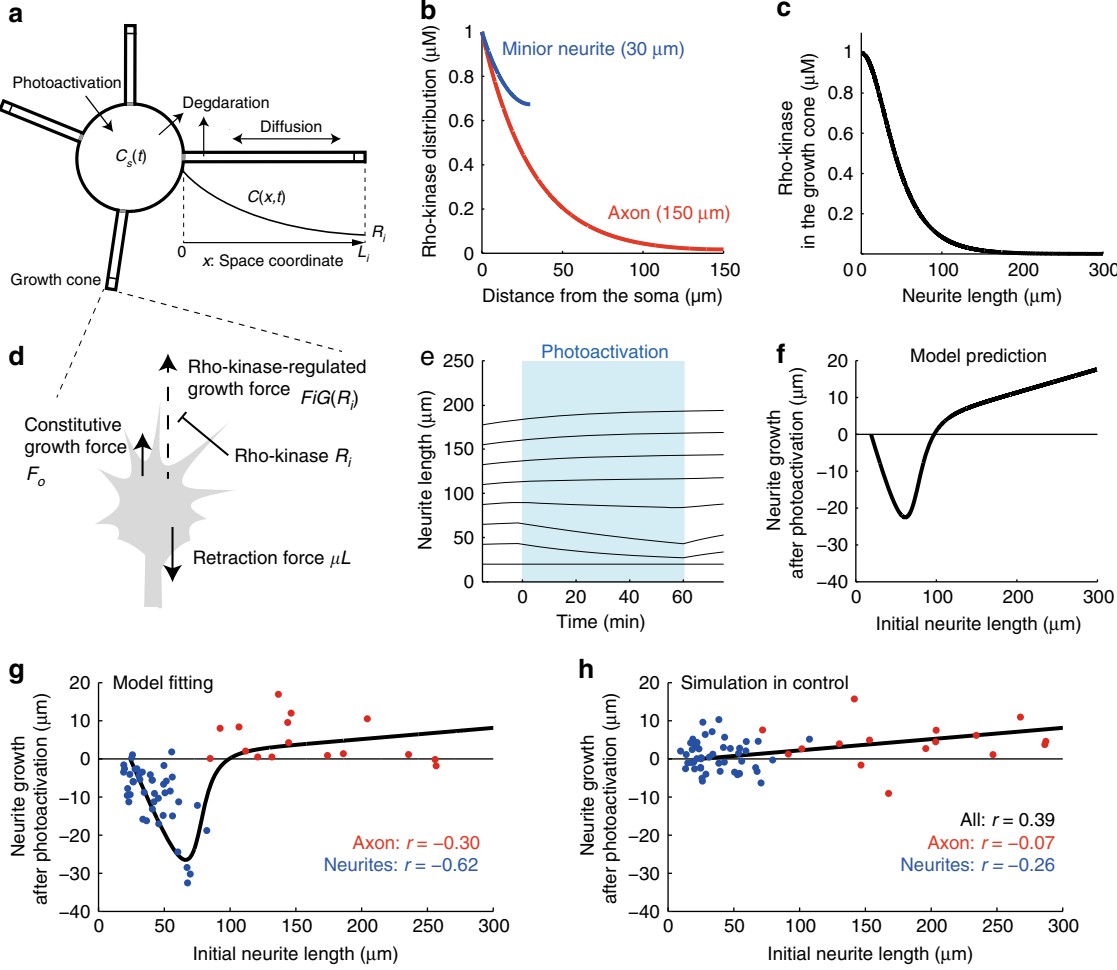

**Fig. 4** Mathematical model for neurite outgrowth regulated by LOVTRAP-Rho-kinase. **a** The model neurite was modeled based on one-dimensional reaction-diffusion of LOVTRAP-Rho-kinase, which was activated in the cell body by illumination, diffused along the neurite, and was then inactivated or degraded. $C_s(t)$, $C_i(x,t)$ and $R_i$ indicate the concentrations of LOVTRAP-Rho-kinase in the cell body at time $t$, along the neurite at $x\,\mu m$ from the neck of neurite $i$ at time $t$, and in the growth cone, respectively. **b** The steady state distribution of LOVTRAP-Rho-kinase along the long axon (*red line*) or the short minor neurites (*blue line*) during photoactivation. Equation (4) was plotted. **c** The concentration of LOVTRAP-Rho-kinase at the tip of the neurite depended on its length, which was mathematically described by Equation (1). **d** Migration of the model growth cone driven by constitutive growth force, Rho-kinase-regulated growth force and retraction force. **e** The simulation dynamics of neurites of various lengths in response to photoactivation of LOVTRAP-Rho-kinase. *Red* and *blue* lines represent typical behaviors of the long axon and short minor neurites, respectively. **f** Model prediction of the relationship between initial neurite length and LOVTRAP-Rho-kinase-dependent neurite retraction. The *black line* was plotted by varying $F_i$, which controls initial neurite length. **g, h** The neurite retraction caused by a 1-h photoactivation of LOVTRAP-Rho-kinase **g** and LOVTRAP-Control **h** was plotted against the initial length of each axon (*red dots*) and minor neurite (*blue dots*). The *black lines* indicate the relationship generated by a simulation in which the parameters ($F_o$, $K/C_o$, **h**, **c**) were adjusted for the best fit to the *red* and *blue* dots. The parameter values used are listed in Methods

degraded (Fig. 4a). Through mathematical analysis, we found that Rho-kinase showed greater accumulation at the tips of short neurites (i.e., minor neurites) compared with long neurites (i.e., an axon) (Fig. 4b). We also derived the concentration of Rho-kinase at the neurite tip under the condition of continuous photoactivation using the following equation:

$$R(L) = C_0 \left[ \cosh\left( \frac{L}{\sqrt{D/k}} \right) \right]^{-1}, \qquad (1)$$

Where $R$, $C_o$, $L_i$, $D$, and $k$ indicate the Rho-kinase concentration at the neurite tip, the Rho-kinase concentration in the cell body during photoactivation, the length of the neurite, the diffusion rate, and the inactivation and degradation rate, respectively. This equation indicates that as the neurite elongates, the Rho-kinase concentration substantially decreases if it is inactivated and degraded (Fig. 4c).

Furthermore, we examined the effect of photoactivated Rho-kinase on neurite outgrowth by developing a mathematical model of Rho-kinase-dependent neurite growth (Fig. 4d) as follows:

$$\frac{\mathrm{d}L_i}{\mathrm{d}t} = -\mu L_i + F_0 + F_i G(R_i), \qquad (2)$$

where $t$, $L_i$, $\mu$, $F_o$, $F_i$, and $R_i$ indicate time, the length of the $i$th neurite, the retraction coefficient, the constitutive growth force shared by all neurites, the growth force of the $i$th neurite and the Rho-kinase concentration at the tip of the $i$th neurite, respectively. $R_i$ was approximately given by Eq. 1 under the assumption that the reaction-diffusion process of the Rho-kinase rapidly reaches quasi-steady state (validity of this approximation was checked by simulating reaction-diffusion of the Rho-kinase along a growing neurite (Supplementary Fig. 5)). The first term represents the retraction force proportional to the neurite

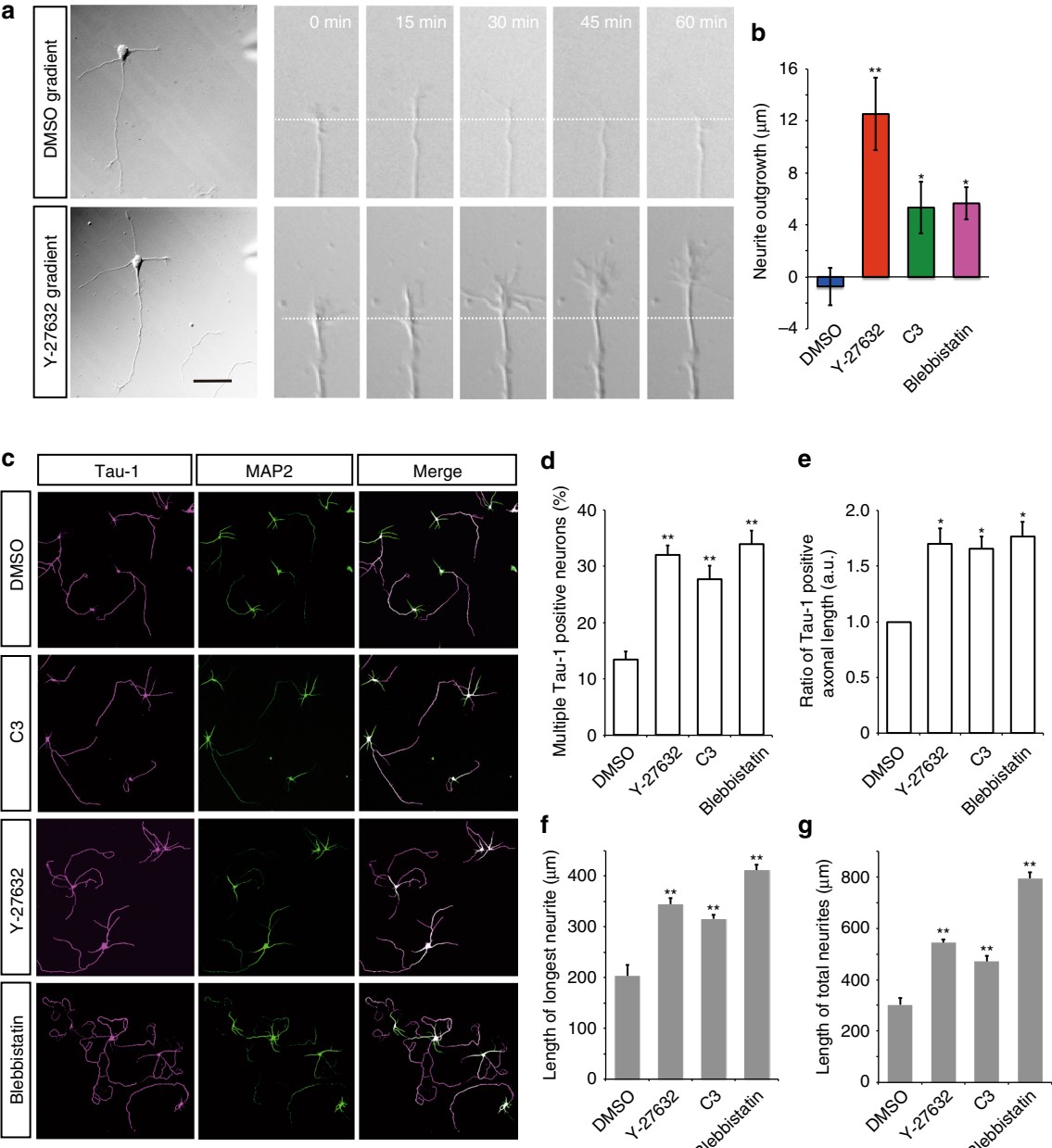

**Fig. 5** RhoA/Rho-kinase was required for the maintenance of neuronal polarity. **a** Local inhibition of Rho-kinase induced minor neurite elongation. DMSO (*top*) or Rho-kinase inhibitor (Y-27632) (*bottom*) was locally applied to a minor neurite of a polarized hippocampal neuron. **b** Local application of Y-27632, C3 or Blebbistatin to a minor neurite. Minor neurite outgrowth was measured (Cont = 17, Y-27632 = 14, C3 = 13, Blebbistatin = 7 neurons from three independent experiments). **c** The effect of the RhoA/Rho-kinase signaling pathway on neuronal polarization. Hippocampal neurons at 3 DIV were treated with Y-27632, C3 or lebbistatin for 48 h. Neurons were co-immunostained at 5 DIV with anti–Tau-1 (*magenta*) and anti-MAP2 (*green*) antibodies. Representative images of neurons are shown. Scale bar, 100 μm. **d** The percentages of neurons with multiple Tau-1-positive axons. **e** Bar graphs indicate the ratio of multiple axonal lengths. **f**, **g** The lengths of the longest neurite **f** and the total neurites **g** were determined (Cont = 91, Y-27632 = 90, C3 = 90, Blebbistatin = 91 neurons from three independent experiments). Error bars represent SEM. *$P < 0.05$ and **$P < 0.01$

length[32], possibly due to membrane tension. In the third term, $G(R_i)$ is a decreasing Hill function that describes inhibitory effect of the Rho-kinase on the growth force $F_i$ via the following: $G(R_i) = K^h/(K^h + R^h)$, where $K$ and $h$ represent the Rho-kinase concentration producing half inhibition and the Hill coefficient describing cooperativity, respectively. Thus, the Rho-kinase-regulated growth force increased in a neurite length-dependent manner as a result of the decrease in Rho-kinase concentration. We assumed that the growth force ($F_i$) was regulated by a set of molecules, e.g., Shootin1 and H-Ras[33, 34], accumulated at the neurite tip through active

transport. In fact, these molecules exhibit greater accumulation at the tips of long neurites compared with short neurites[33, 34]. Consistently, in our model, the longer neurite results from larger growth force.

By simulating the dynamics of neurites in response to the photoactivation of Rho-kinase, we produced a growth-discouraging effect on short neurites and no effect on long neurites (Fig. 4e), consistent with Fig. 3g, h, respectively. Moreover, this model predicted a U-shaped relationship between neurite length at the onset of photoactivation and neurite growth length after photoactivation (Fig. 4f).

To confirm this model prediction, we investigated the relationship between the retraction induced by photoactivation and the initial length of each axon and minor neurite. We found that Rho-kinase-dependent growth-discouraging behavior was increased in minor neurites that were within 75 µm of the cell body but not in axons over 100 µm away from the cell body (blue and red dots in Fig. 4g), which agrees qualitatively with the model prediction (Fig. 4f). We estimated the model parameters to fit the relationship by means of a non-linear regression method. The model with the estimated parameters exhibited excellent quantitative agreement with the experimental data (black line in Fig. 4g). Using the estimated parameters, we, furthermore, simulated the control condition, in which the Rho-kinase is not released by photoactivation. This model then reproduced a positive correlation between neurite length at the onset of photoactivation and neurite growth length after photoactivation (Fig. 4h). These results suggest that active Rho-kinase diffuses from the cell body to short minor neurites but not to long axons and subsequently retracts the growing minor neurites.

**RhoA/Rho-kinase regulates single axon formation**. Based on the above results, we speculated that the presence of active Rho-kinase in minor neurites prevents their outgrowth to ensure single axon formation. To investigate the spatial distribution of endogenous Rho-kinase, we immunostained locally stimulated neurons with anti-Rho-kinase[35]. The 2.5-D reconstruction analysis showed that, in stage 3 hippocampal neurons, Rho-kinase was localized throughout the cytoplasm but prominently found in minor neurites and the proximal region of the axon (Supplementary Fig. 6a). Interestingly, when NT-3 was locally applied to the axon, Rho-kinase was enriched in the cell body and minor neurites compared with the axon (Supplementary Fig. 6a), which is similar to the results of the Rho-kinase diffusion model (Fig. 4c). To examine whether local inhibition of Rho-kinase induces minor neurite elongation, we locally exposed the growth cone of a minor neurite of a stage 3 neuron to Y-27632 (Fig. 5a, left panel). Local application of Y-27632 to the minor neurite induced rapid neurite elongation in a time-dependent manner (Fig. 5a, b, Supplementary Movie 9 and Supplementary Movie 10). In addition, local application of C3 or blebbistatin also induced neurite elongation (Fig. 5a, b), indicating that RhoA/Rho-kinase inhibits minor neurite outgrowth

presumably through local contractility. To determine whether RhoA/Rho-kinase prevents minor neurites from eventually developing into axons, hippocampal neurons were treated with Y-27632, C3 or blebbistatin at 3 DIV and then subjected to immunostaining with anti-Tau-1 and anti-MAP2 at 5 DIV. All of these treatments increased the number of neurons with multiple Tau-1-positive axons (Fig. 5c, d). Notably, these Tau-1-positive newly formed axons grew from the distal tips of the minor neurites of polarized neurons (Supplementary Fig. 6b). The length of multiple axons, the longest neurite and the total number of neurites also increased in response to the treatments (Fig. 5e–g). These results clearly suggest that the local activity of Rho-kinase at minor neurites is necessary to generate single axon formation because it directly prevents multiple axon formation.

**GEF-H1 is a novel substrate of CaMKI in neurons**. We recently developed a novel phospho-proteomic method called kinase-interacting substrate screening (KISS) that can comprehensively identify several kinase substrates[36]. We used this method to identify the CaMKI substrates that lead to RhoA activation. Brain extracts were applied to affinity beads coated with the catalytic domain of CaMKI to form the kinase-substrate complex with or without ATP and $Mg^{2+}$ (Supplementary Fig. 7a). The samples were digested with trypsin and subjected to liquid chromatography/tandem mass spectrometry (LC/MS/MS). A total of 1142 phosphorylation sites of 498 phospho-proteins were identified. Consistent with previous reports, sequence alignment of the predicted phosphorylation sites showed that CaMKI phosphorylated [R]-X-X-[S/T] sequences (Supplementary Fig. 7b). Notably, numerous proteins were detected, including small GTPase regulators (GEF/GAP), cytoskeletal proteins, kinases, membrane proteins, and transcriptional factors (Table 1). In this study, we focused on guanine nucleotide exchange factor-H1(GEF-H1) as a candidate substrate of CaMKI because GEF-H1 is the only RhoA-specific regulator among these phospho-proteins[37–39]. To confirm the phospho-proteomic results, we performed an in vitro kinase assay with GEF-H1 fragments with an N-terminal-containing C1 domain (1-84 aa), an M/1 region (85-130 aa), an M/2 region (131-227 aa), a DH/PH domain (228-580 aa), and a CC domain (581-985 aa) (Fig. 6a). We found that CaMKI phosphorylated the GST-GEF-H1-M/1 regions but not the other fragments (Fig. 6b). To identify the phosphorylation

**Table 1 Candidate substrates of CaMKI in neurons**

| Functions | Candidate substrates |
|---|---|
| GEFs | Arhgef2/GEF-H1, Arfgef3, Ephexin-1 |
| GAPs | Agap2, Arfgap1, Git1, Ranbp1 |
| Cytoskeletal proteins | Add1, Add2, Camsap2, Capza2, Capzb, Cmya3, Coro7, Crmp1, Crmp3, Crmp4 |
| | Dbnl, Dnm, Enah, Epb4.1, Epb4.1l1, Epb4.9, Evl, Gfap, Lasp1, Map1a |
| | Map1b, Map2, Map4, Map6, Map6d1, Mapre3, Mapt, Nef3, Nir2, Pacsin1 |
| | Shroom2, Tppp, Tuba1, Tuba1b, Tuba3, Tuba4, Tuba8, Tubb2a, Tubb2c, Tubb3, Tubb4, Tubb5 |
| Protein kinases | Aak1, Akt3, Araf, Camk1, Camk1g, Camk2b, Camk2d, Camkk1, Camkk2, Camkv |
| | Cdk18, Cdk5, Erk1, Erk2, Fak2, Gsk3a, Gsk3b, Hk1, Itpk1, Itpka |
| | Map2k4, Map3k7, Mark1, Mark2, MST3, Mylk, Pak1, Pak2, Pak7 |
| | Pctk1, Pfk-C, Pfkfb2, Pfkl, Pfkm, Pkcb, Pkcc, Pkce, Pkm2, Prkar1a, Prkar2b |
| | Prkca, Prkra, Rps6ka2, Slk, Stk38l, Tbckl |
| Scaffold proteins | Akap12, Alp, Homer, Shank2, Shank3 |
| Adapter proteins | Nck1, Ap2m1 |
| Moter proteins | Dctn2, Dctn4, Dhc1, Dnci1, Dncli2, Khc, Klc2 |
| Membrane proteins | Arf6, Ap1gbp1, Apba1, Atg16l1, Epn1, Hid1, Dlp1, Necap1, Ra4l0, Rtn3 |
| | Sec14l1, Sec16a, Sec31a, Sh3gl1, Snca, Snx1, Snx3, Snx5, Snx6, Snx17 |
| | Tbc1d10b_predicted, Tsga14, Unc13c, Vps33b, Vps52, Vps53 |
| Transcription factors | Ctbp2, Ccar1, Crtc1, Evi1, Gtf2i, Hnrpdl, Rrn3 |

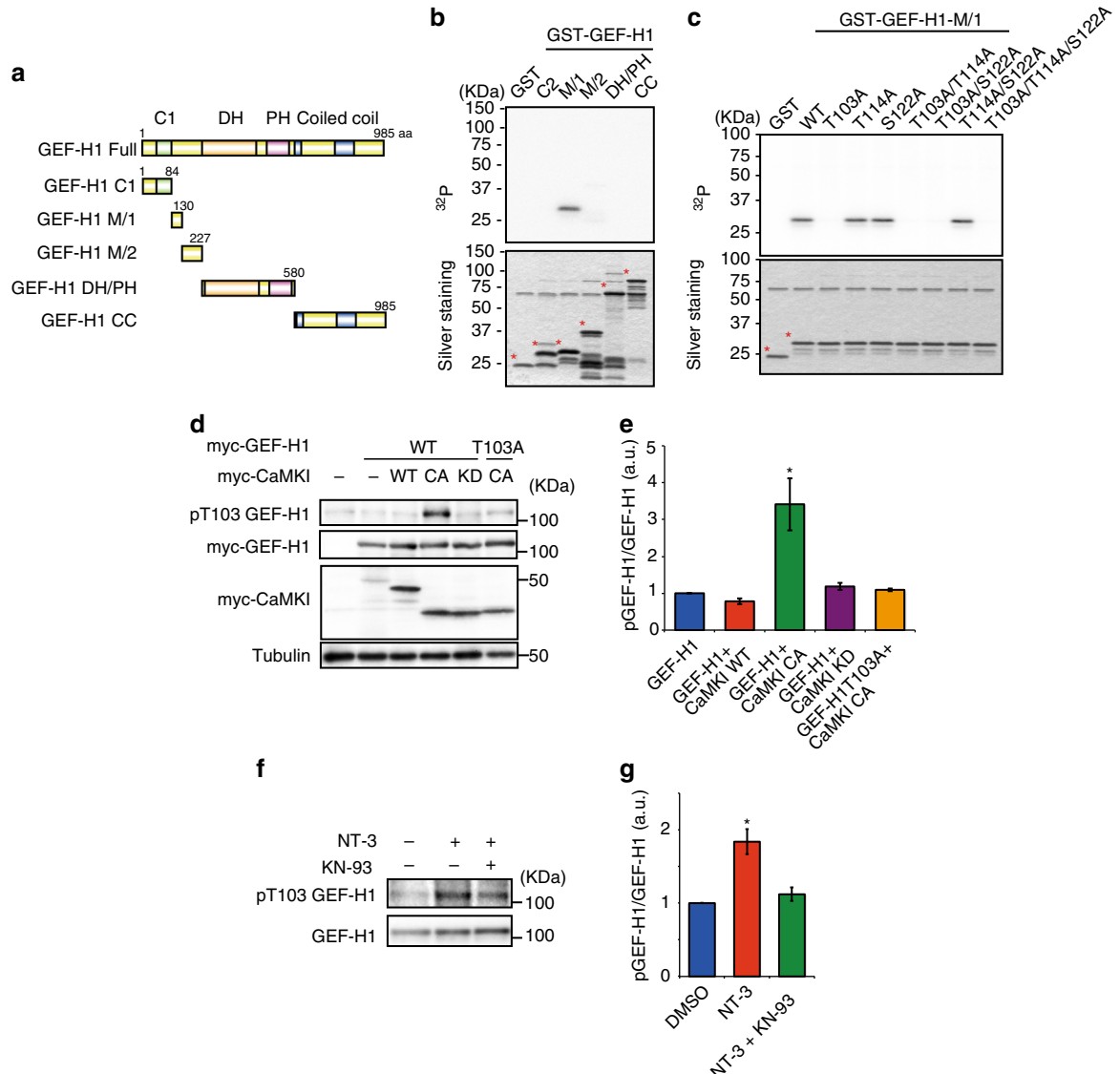

**Fig. 6** GEF-H1 is identified as a new substrate of CaMKI in the brain. **a** The domain structures of GEF-H1 and its various fragments are represented. **b** Direct phosphorylation of GEF-H1 by CaMKI. Each purified fragment of GEF-H1 was incubated with recombinant CaMKI–cat in the presence of [γ-$^{32}$P]ATP in vitro. *Asterisks* indicate intact GST-fusion proteins. **c** Phosphorylation of GEF-H1 at Thr103 by CaMKI. Each purified Ala mutant of GEF-H1 was incubated with recombinant CaMKI–cat in the presence of [γ-$^{32}$P]ATP in vitro. *Asterisks* indicate intact GST-fusion proteins. **d**, **e** Phosphorylation of GEF-H1 in COS-7 cells. GEF-H1 was co-transfected with myc-CaMKI-WT, -constitutive active form (CA) or -kinase-dead form (KD) into COS-7 cells. Cell lysates were analyzed by immunoblotting with anti-pT103, anti-myc, and anti-tubulin antibodies. **f**, **g** Phosphorylation of endogenous GEF-H1 in hippocampal neurons. Hippocampal neurons were treated with NT-3 with or without KN-93. Cell lysates were analyzed by immunoblotting with anti-pT103 and anti-GEF-H1 antibodies. Error bars represent SEM. *$P < 0.05$ and **$P < 0.01$

sites, we prepared GEF-H1-M/1 point mutants (mutated to Ala) with potential CaMKI phosphorylation sites in this region (Thr103, Thr114, and Ser122). Ala substitutions at Thr103 (T103A, T103A/S144A, T103A/S122A, and T103A/T114A/S122A) abolished phosphorylation by CaMKI, but the other Ala mutants had no effect (Fig. 6c), suggesting that CaMKI directly phosphorylates GEF-H1 at Thr103.

To verify the phosphorylation of GEF-H1 in vivo, we attempted to generate a phospho-specific antibody against Thr103, and we successfully produced the anti-pT103 antibody. In immunoblotting analysis, the anti-pT103 antibody detected GEF-H1 phosphorylation when co-expressed with a constitutively active form of CaMKI (CaMKI-CA) but not CaMKI-WT or the kinase-dead form of CaMKI (CaMKI-KD) in COS-7 cells (Fig. 6d, e). In contrast, GEF-H1

phosphorylation was not detected in the T103A mutant of GEF-H1, despite coexpression with CaMKI-CA (Fig. 6d, e). We next explored GEF-H1 phosphorylation under more realistic physiological conditions. The stimulation of hippocampal neurons with NT-3 significantly increased the phosphorylation of endogenous GEF-H1 at Thr103 (Fig. 6f, g). The NT-3-induced phosphorylation of GEF-H1 was markedly reduced by KN-93 (Fig. 6f, g). We next investigated the spatial distribution of phosphorylated GEF-H1 at Thr103 after local application of NT-3. The cell bodies of the NT-3-stimulated neurons showed significantly elevated amounts of phosphorylated GEF-H1, whereas the axons and minor neurites did not (Supplementary Fig. 7c–f). These results suggest that NT-3 induces the phosphorylation of GEF-H1 at Thr103 via CaMKI in hippocampal neurons.

**Phosphorylation of GEF-H1 regulates single axon formation.** To examine the involvement of GEF-H1 in NT-3-induced RhoA activation and minor neurite retraction, we prepared short interfering RNAs against GEF-H1 (siGEF-H1). Knockdown efficiency was tested by immunoblotting, and the results demonstrated that siGEF-H1#2 was more effective (Supplementary Fig. 8a). Knockdown of GEF-H1 by siGEF-H1#2 abolished

minor neurite retraction and the increment in RhoA activity in the cell body triggered by NT-3 (Fig. 7a, b). Of note, knockdown of GEF-H1 reduced RhoA activity even in control neurons (Fig. 7b). These results suggest that GEF-H1 is required for NT-3-induced minor neurite retraction and RhoA activation.

The Thr103 site of GEF-H1 is located close to the N-terminal C1 domain, which regulates the activity of GEF-H1[37].

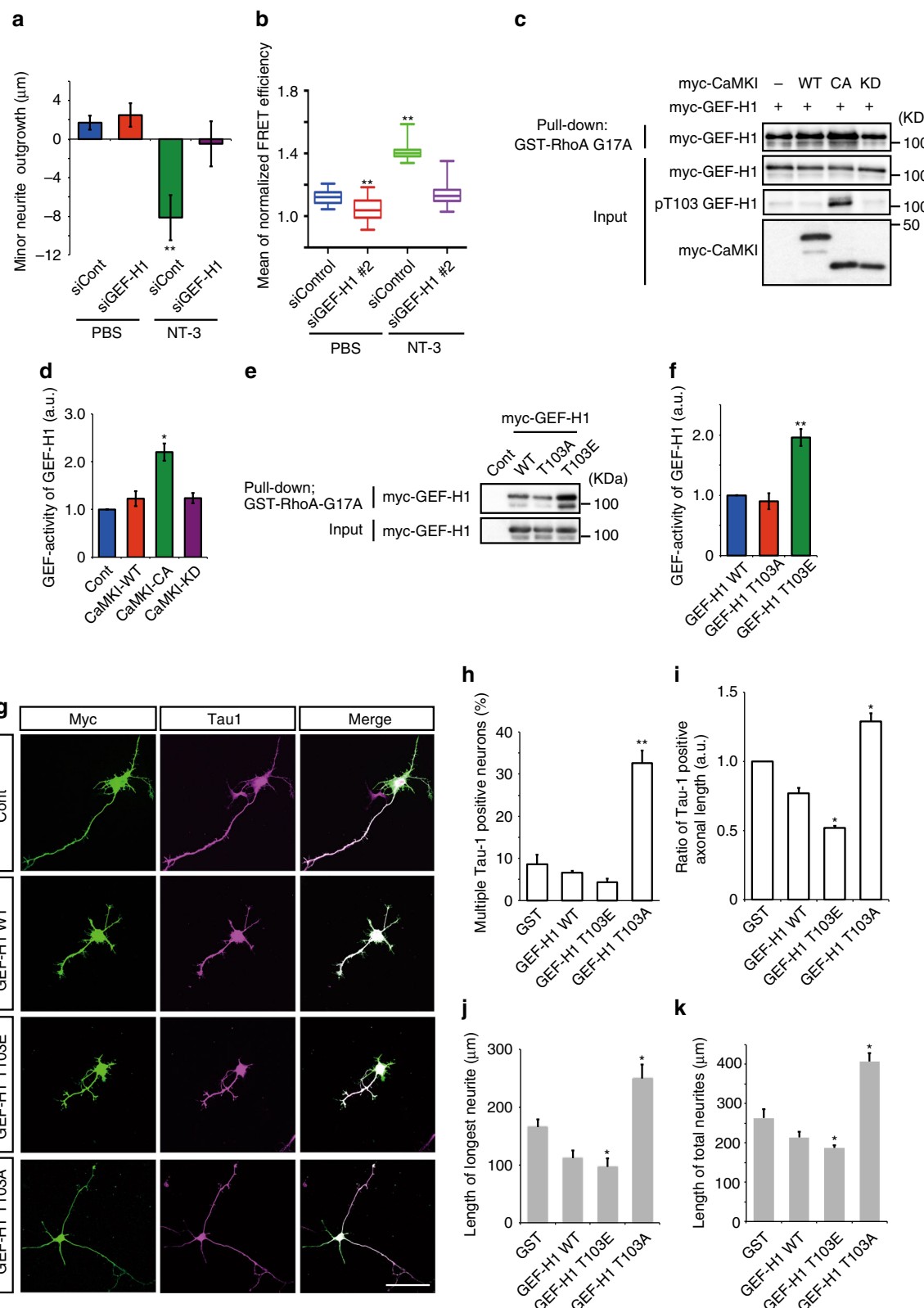

To investigate whether the phosphorylation of GEF-H1 affects its GEF activity, we utilized beads coated with a nucleotide-free mutant of RhoA (RhoA-G17A) and performed an affinity precipitation assay to specifically pull down activated GEF-H1[40]. Coexpression of CaMKI-CA, but not of CaMKI–WT or CaMKI–KD, dramatically increased the amount of myc-GEF-H1 that precipitated with RhoA-G17A and stimulated the phosphorylation of GEF-H1 at Thr103 (Fig. 7c, d). In addition, we examined whether phosphorylation of GEF-H1 at Thr103 by CaMKI enhanced its GEF activity using two GEF-H1 mutants: T103A, which is constitutively unphosphorylated, and T103E, which mimics the phosphorylated form. The expression of GEF-H1–T103E significantly increased the amount of myc-GEF-H1 that precipitated with RhoA-G17A compared with GEF-H1–WT or –T103A (Fig. 7e, f). These results suggest that CaMKI phosphorylates GEF-H1 and activates its GEF activity on RhoA.

We next examined the effect of GEF-H1 phosphorylation by CaMKI on neuronal polarization. We transfected myc-GEF-H1–WT, –T103E or –T103A into hippocampal neurons at 3 DIV and then subjected to immunostaining at 4 DIV. We found that expression of GEF-H1–T103A, but not of GEF-H1–WT or GEF-H1–T103E, induced multiple axon formation (Fig. 7g–k). Furthermore, hippocampal neurons expressing GEF-H1–WT and GEF-H1–T103E had shorter axons and minor neurites compared with control neurons, whereas neurons expressing GEF-H1–T103A had significantly increased axon and total neurite lengths (Fig. 7i–k).

We next examined the effect of GEF-H1 knockdown on neuronal polarization. We prepared pSico-mCherry-GEF-H1 (shGEF-H1), which allowed the expression of short hairpin RNA (shRNA) against GEF-H1 under the control of Cre recombinase[12, 41]. Transfection of shGEF-H1#1 and shGEF-H1#2 with Cre effectively decreased the expression level of GEF-H1 in Neuro2a cells and hippocampal neurons (Supplementary Fig. 8b, c). Knockdown of GEF-H1 at 3 DIV induced the formation of multiple Tau-1-positive axons in hippocampal neurons (Supplementary Fig. 8d, g) and increased the longest neurite length and total neurite length compared with control neurons (Supplementary Fig. 8h, i). We next performed rescue experiments in GEF-H1-depleted neurons with shRNA-resistant forms of GEF-H1 (GEF-H1 Res), which restored the expression of GEF-H1 in Neuro2a cells (Supplementary Fig. 8e). In hippocampal neurons, expression of GEF-H1–WT Res or –T103E Res in GEF-H1-depleted neurons reversed the knockdown effects by leading to the formation of multiple axons and increasing axon and total neurite lengths to the levels observed in control neurons (Supplementary Fig. 8f–i). In contrast, GEF-H1–T103A Res did not recover the knockdown effects (Supplementary Fig. 8f–i). These results suggest that GEF-H1 and its phosphorylation at Thr103 are important for single axon formation in hippocampal neurons.

**Phosphorylation of GEF-H1 regulates polarity in vivo.** We further investigated the role of GEF-H1 in neuronal polarization in vivo using an in utero electroporation system to introduce GEF-H1 and shRNA into pyramidal neurons[41, 42]. Pyramidal neurons first extend multiple minor processes and are called multipolar (MP) cells in intermediate zone (IZ). These MP cells subsequently transform into bipolar (BP) cells with a trailing process and a leading process, which finally develop into an axon and a dendrite, respectively[2, 3]. BP cells are completely polarized and migrate toward the cortical plate (CP). We transfected the E13.5 neocortex with pTα-LPL, pTα-LPL-GEF-H1–WT, pTα-LPL-GEF-H1–T103E, or pTα-LPL-GEF-H1–T103A, together with pTα-Cre and pTα-LPL-Lyn-EGFP. The majority of the control cells had entered into the CP, whereas most of the GEF-H1–WT-, –T103E-, or –T103A-expressing pyramidal cells failed to acquire a BP morphology and migrate into the (Fig. 8a–e). Notably, the GEF-H1–T103E-expressing pyramidal cells exhibited an abnormal round morphology, whereas the GEF-H1–T103A-expressing pyramidal cells showed impaired formation of leading processes (Fig. 8a, d, e). We also found that most of the shGEF-H1-expressing pyramidal cells failed to acquire a BP morphology and enter the CP (Fig. 8f–j). The knockdown phenotype was rescued by coelectroporation with GEF-H1–WT Res but not with –T103E Res or –T103A Res (Fig. 8f–j). GEF-H1–T103E Res-expressing pyramidal cells did not show restored trailing process formation (Fig. 8f, i, j). In contrast, GEF-H1–T103A Res-expressing pyramidal cells did not show restored leading process formation (Fig. 8f, i, j). These findings suggest that GEF-H1 and its phosphorylation at Thr103 are important for neuronal polarization in the developing neocortex.

## Discussion

We succeeded in identifying a long-range inhibitory signaling pathway that is enhanced by the local application of NT-3 onto a nascent axon (Supplementary Fig. 9). Because this signaling is transient, identifying the inhibitory machinery posed substantial technical challenges[43, 44], as the $Ca^{2+}$ waves could only be occasionally and transiently observed in axons under static conditions (Fig. 2 and Supplementary Movie 1). In the absence of exogenous neurotrophins, cultured neurons stochastically extend one axon and other minor neurites that never develop into axons (the stochastic model)[1–5, 44–48]. Thus, typical neurons could possess long-range inhibitory signaling for a single axon formation without the need for exogenous factors. Consistently, the attenuation of long-range $Ca^{2+}$ propagations from the axon to the cell body and the subsequent CaMKI activity in the cell body induced minor neurite elongation (Fig. 2 and Supplementary Fig. 2). Moreover, local inhibition of Rho-kinase in minor neurites in stage 3 neurons induced rapid neurite elongation and

**Fig. 7** Phosphorylation of GEF-H1 by CaMKI increased its GEF activity. **a** NT-3-induced minor neurite retraction was abolished by knockdown of GEF-H1 (siCont/PBS = 29, siCont/NT-3 = 26, siGEF-H1#2/PBS = 23, siGEF-H1#2/NT-3 = 26 neurites from three independent experiments). **b** The increase in FRET efficiency in the cell body induced by NT-3 was abolished by knockdown of GEF-H1 (siCont/PBS = 5, siCont/NT-3 = 8, siGEF-H1#2/PBS = 6, siGEF-H1#2/NT-3 = 15 neurons from three independent experiments). **c** CaMKI increased the GEF activity of GEF-H1. GEF-H1 was co-transfected with myc-CaMKI-WT, -CA or –KD into COS-7 cells. The cell extracts were incubated with GST-RhoA-G17A-bound beads. Active GEF-H1 was pulled down and detected by immunoblotting with anti-myc (top). Total GEF-H1 and CaMKI were detected by immunoblotting with anti-myc (*middle* and *bottom*, respectively). **d** Bar graphs indicate the ratio of GEF-H1 activity. **e** The phospho-mimic mutant GEF-H1 exhibited an increment in GEF activity. Cell lysates expressing GEF-H1–WT, –T103A or –T103E were incubated with GST-RhoA-G17A-bound beads. Active (*top*) or total (*bottom*) GEF-H1 was detected by immunoblotting with an anti-myc antibody. **f** Bar graphs indicate the ratio of GEF-H1 activity. **g** myc-GST (Cont), myc-GEF-H1–WT, –T103A or –T103E was transfected into neurons at 3 DIV. Representative images of the neurons at 4 DIV are shown. Scale bars represent 100 μm. **h** The percentages of neurons with multiple Tau-1-positive axons. **i** Bar graphs indicate the ratio of multiple axonal lengths. **j, k** The length of the longest neurite **i** and the total neurite length **j** were determined (GST = 86, GEF-H1 WT = 90, GEF-H1 T103E = 88, GEF-H1 T103A = 90 neurons from three independent experiments). Error bars represent SEM. *$P < 0.05$ and **$P < 0.01$

subsequent multiple axon formation (Fig. 5). These results clearly indicate that long-range inhibitory signaling plays a critical role in neuronal polarization not only in the NT-3-stimulated model but also in the stochastic model.

CaMKI plays a crucial role in the establishment of neuronal polarity[12, 19, 20]. However, the substrates used by CaMKI during neuronal polarization have not yet been identified. In this study, we found that CaMKI phosphorylated GEF-H1 at Thr103, which is located close to the C1 domain. The phosphorylation of GEF-H1 by CaMKI enhanced its GEF activity and suppressed axon formation in vitro and in vivo. GEF-H1 is associated with microtubules through the C1 domain and is activated following

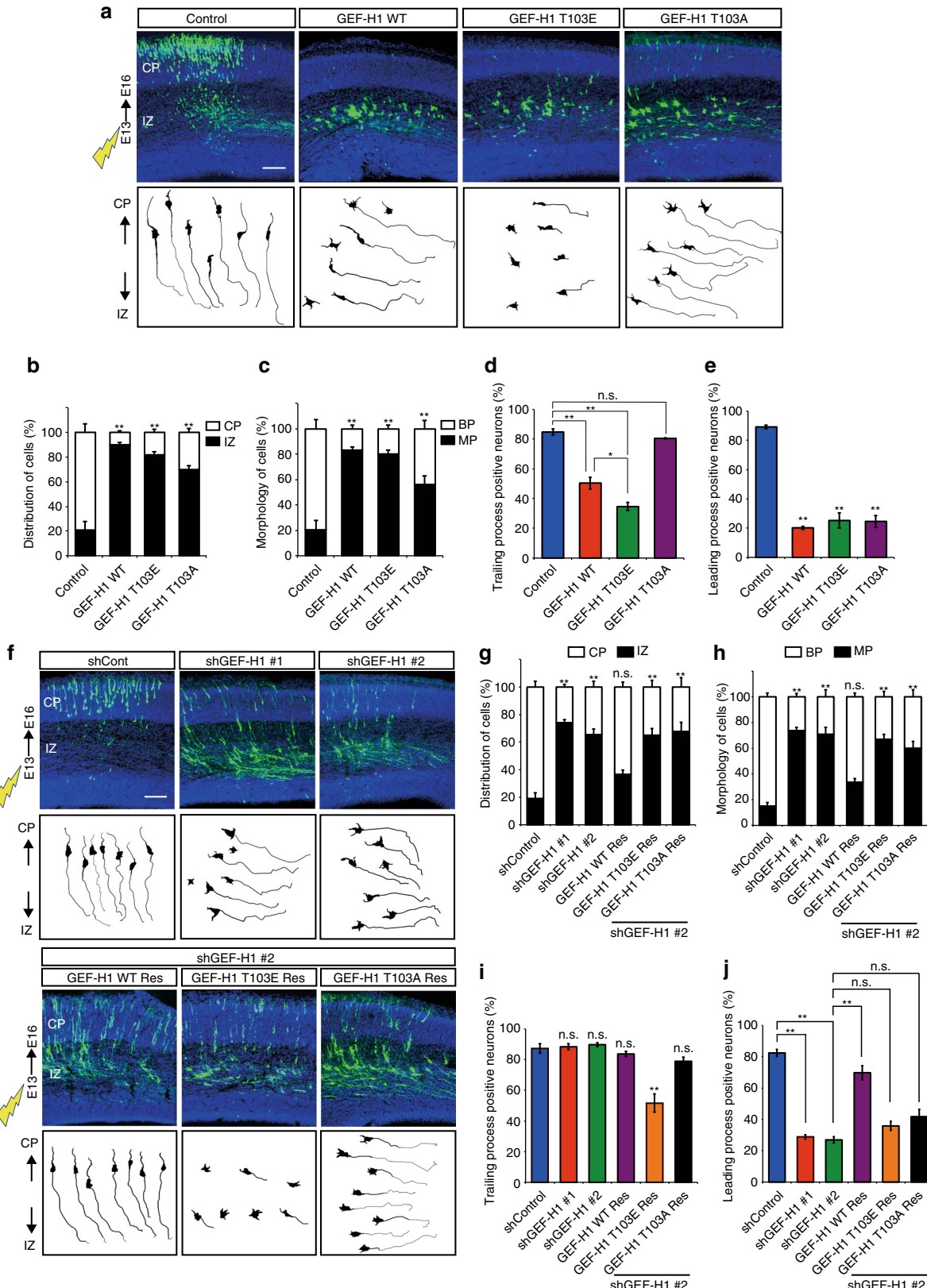

its release from microtubules[37, 38]. In addition, several GEFs, such as Vav2 and Tiam1, contain regulatory domains that inhibit activity through an intramolecular interaction, and the dissociation of this interaction through a conformational change leads to GEF activation[49, 50]. The GEF-H1 phosphorylated by CaMKI may be activated by its release from microtubules and/or by the halting of auto-inhibition. On the other hand, the non-phosphorylation mutant of GEF-H1 induced the formation of multiple axons (Fig. 7), indicating that GEF-H1 maintains single axon formation depending on its phosphorylation state. However, we cannot exclude the possibility that the non-phosphorylation mutant of GEF-H1 would have functioned as a dominant-negative factor.

The subcellular polarity of the small GTPases plays a critical role in cell polarization[23–27]. We found that NT-3-induced Ca$^{2+}$ waves led to the polarized activation of RhoA/Rho-kinase in the cell bodies of hippocampal neurons. Importantly, GEF-H1 knockdown fully inhibited the polarized activation of RhoA, indicating that GEF-H1 mediates this polarized activation. The mechanism by which GEF-H1 regulates polarized RhoA activity in neurons remains unknown. One possibility is that the polarized activity of RhoA results from the difference in phosphorylation states of GEF-H1 in neurons. Indeed, phosphorylation of GEF-H1 at Thr103 was increased by NT-3 in the cell body but not in the axon or minor neurites (Supplementary Fig. 7). Another possibility is that GEF-H1 may be specifically inactivated in the axon by PKA because PKA has been activated in the growing axon by neurotrophins[51, 52]. RhoA/Rho-kinase activity in the cell body is important for preventing the formation of unnecessary axons. We found that photo-activation of RhoA or Rho-kinase in the cell body resulted in rapid minor neurite retraction but not axon retraction. Additionally, our computational modeling helps explain how active Rho-kinase diffuses from the cell body to short minor neurites but not to long axons and subsequently retracts growing minor neurites to prevent multiple axon formation.

Rho-kinase governs a wide variety of cellular events such as actin filament and microtubule dynamics that regulate axon formation[5, 46, 51] through several of its substrates, including LIM kinase, CRMP-2 and Tau[53–55]. In addition, local inhibition of myosin II induced minor neurite elongation to some extent, indicating that the role of Rho-kinase in neuronal polarization is partly mediated by contractility. Interestingly, local stimulation with NT-3 induced continuous activation of RhoA in minor neurites (Supplementary Fig. 3). We previously reported that Rho-kinase phosphorylates and inactivates p190RhoGAP, thereby leading to sustained activation of RhoA[56]. Rho-kinase also inactivates Rac1 through phosphorylation of Par3 and Rac GEFs (STEF/Tiam)[1, 3, 57]. Following Rho-kinase activation by long-range inhibitory signaling, the activated RhoA may be sustained and diffused by Rho-kinase-dependent inactivation of

RhoGAP, ultimately guaranteeing neuronal polarity through multiple pathways in minor neurites.

We previously reported that CaMKI and RhoA/Rho-kinase regulates MP-to-BP transition and proper neuronal entry into CP[12, 27]. However, the regulatory interaction between CaMKI and RhoA/Rho-kinase during neuronal polarization in vivo remains undetermined. Here, we focused on GEF-H1 functions and found that GEF-H1 and its phosphorylation were required for the MP-to-BP transition and the subsequent neuronal migration (Fig. 8). The expression of GEF-H1-T103E impaired MP-to-BP transition in vivo, which was consistent with previous results of active CaMKI and RhoA[12, 58]. We recently found that expression of the dominant-negative RhoA or Rho-kinase also impairs MP-to-BP transition and neuronal migration[27]. The appropriate activity of RhoA/Rho-kinase is vitally essential for neuronal polarization, and, therefore, might be properly regulated by CaMKI-mediated phosphorylation of GEF-H1 in the developing neocortex. Interestingly, GEF-H1-T103A-expressing pyramidal cells formed unidentified multiple processes, which were shorter than trailing processes, resulting in an interruption of the leading process (future dendrite) formation in vivo, which was similar to our in vitro findings. Thus, Ca$^{2+}$/CaMKI/GEF-H1/RhoA/Rho-kinase signaling is a novel signaling pathway that function in long-range inhibition to establish and maintain neuronal polarity during neuronal development.

## Methods

**Plasmid construction.** Mouse GEF-H1 was amplified by PCR and subcloned into pCAGGS and pTα-LPL vectors. The mutants of GEF-H1-T103A, GEF-H1-T114A, GEF-H1-S122A, GEF-H1-T103A/T114A, GEF-H1-T103A/S122A, GEF-H1-T114A/S122A, GEF-H1-T103A/T114A/S122A, and GEF-H1-T103E were generated using a site-directed mutagenesis kit (Stratagene) (Supplementary Table 1). cDNA fragments encoding GEF-H1 mutants (GEF-H1 C1, GEF-H1 M/1, GEF-H1 M/2, GEF-H1 DH/PH, and GEF-H1 CC) were subcloned into pGEX vector (GE Healthcare) (Supplementary Table 1). GST-tagged proteins were produced in BL21 (DE3) *Escherichia coli* cells and purified on glutathione–Sepharose 4B beads (GE Healthcare). The cDNAs encoding wild-type CaMKIa, a constitutively active cytosolic form of CaMKIa (CaMKIa-293NES), and a kinase-dead form of CaMKIa (CaMKIa-293NES-KA) were obtained as previouslydescribed[12, 20]. pCAGGS-Raichu-RhoA-CR was a gift from Michael Lin as previously described (Stanford University, CA) (plasmid #40258, Addgene)[27, 28]. NTOM20-mVenus-LOV2 was obtained as previously described[30]. cDNAs encoding mCherry-Zdk1, mCherry-Zdk1-RhoA Q63L, mCherry-Rho-kinase CAT-Zdk1, and mCherry-Rho-kinase CAT KD-Zdk1 were subcloned into pTrix and pCAGGS vectors. The target sequences for siGEF-H1#1 and siGEF-H1#2 were as follows: 5′-CUGAUU-GUCCUUACCAGUUTT-3′ and 5′-GAAGCAAGAUGUCAUCUAUUTT-3′, respectively. pSico-mCherry was generated by inserting mCherry in place of EGFP as previously described[12]. The mouse GEF-H1 shRNA plasmid was generated by inserting the annealed oligonucleotides into apSicovector[59, 60]. The target sequences for shGEF-H1 #1 and shGEF-H1#2 were as follows: 5′-GGGCTGCGGTTGCTTCTGTAA-3′ and 5′-GGGATGCTGGAA-GAGTTGCAG-3′, respectively[24]. shRNA-resistant GEF-H1 was generated by changing the targeted sequence of shGEF-H1 #2 to the sequence 5′-GGAATGC-TAGAAGAATTGCAG-3′ by PCR-based site-directed mutagenesis. All constructs were confirmed by DNA sequencing. pEF-BOS-GST-RhoA-G17A was used for the pull-down assay[40, 61].

---

**Fig. 8** The role of GEF-H1 in neuronal polarization in vivo **a** pTα-LPL-Lyn-EGFP was coelectroporated with Tα-Cre and pTα-LPL (control), pTα-LPL-GEF-H1–WT, –T103E, or –T103A into cerebral cortices at E13 followed by fixation at E16. Coronal sections were prepared and immunostained with an anti-EGFP antibody (green). Nuclei were stained with Hoechst 33342 (blue). The tracing images depict EGFP-labeled cells (bottom panels). Scale bars, 100 μm. **b** Quantification of the distributions of EGFP-positive cells in distinct regions of the cerebral cortex (CP and IZ). **c** Percentages of EGFP-positive cells with bipolar (BP) or multipolar (MP) morphologies in the cerebral cortex. **d, e** Percentages of trailing process-positive **d** or leading process-positive **e** neurons (pTα-LPL-EGFP = 6, pTα-LPL-GEF-H1-WT = 9, pTα-LPL-GEF-H1-T103E = 9, pTα-LPL-GEF-H1-T103A = 4 brains from three independent experiments). **f** pSico-mCherry-shGEF-H1 #1 or pSico-mCherry-shGEF-H1 #2 was coelectroporated with Tα-Cre and Tα-LPL-Lyn-EGFP or shRNA-resistant forms of GEF-H1 into cerebral cortices at E13 followed by fixation at E16. Coronal sections were prepared and immunostained with an anti-EGFP antibody (green). Nuclei were stained with Hoechst 33342 (blue). The tracing images showed the EGFP-labeled cells (bottom panels). Scale bars represent 100 μm. **g** Quantification of the distributions of EGFP-positive cells in distinct regions of the cerebral cortex (CP and IZ). **h** Percentages of EGFP-positive cells with bipolar (BP) or multipolar (MP) morphologies in the cerebral cortex. **i, j** Percentages of trailing process-positive **i** or leading process-positive **j** neurons (pSico-mCherry = 4, pSico-mCherry-shGEF-H1 #1 = 8, pSico-mCherry-shGEF-H1 #2 = 4, pTα-LPL-GEF-H1-WT Res = 5, pTα-LPL-GEF-H1-T103E Res = 13, pTα-LPL-GEF-H1-T103A Res = 10 brains from three independent experiments). Error bars represent SEM. *$P < 0.05$ and **$P < 0.01$

**Antibodies**. The following antibodies and materials were used: a monoclonal rabbit anti-GEF-H1 antibody (1:1000, Cell Signaling, 4076); a polyclonal rabbit anti-c-myc antibody (1:1000, Santa Cruz Biotechnology, sc-789); a monoclonal mouse anti-myc antibody (9E10, 1:1000, sc-40), a monoclonal mouse anti-α-tubulin antibody (DM1A, 1:2000, T9026), a monoclonal mouse anti-GST antibody (1:1000, Sigma, G 1160), a monoclonal mouse anti-Tau-1 antibody, (1:2000, Millipore Bioscience Research Reagents, MAB3420) a polyclonal anti-MAP2 antibody (1:1000, Chemicon, AB5622), monoclonal mouse anti-class III β-tubulin and polyclonal rabbit anti-class III β-tubulin antibodies (Tuj1, 1:2000, Covance, MMS-435P), a monoclonal mouse anti-GFP antibody (1:1000, Roche Diagnostics, 11814460001), a polyclonal rabbit antibody against GEF-H1 phosphorylated at Thr-103 was produced against the phosphopeptide (Cys-Leu$^{98}$-Leu-Arg-Asn-Asn-phospho-Thr$^{103}$-Ala-Leu-Gln-Ser-Val-Ser$^{109}$) by Wako, a polyclonal anti-NT-3 anti body (1:500, Promega, G165A), a monoclonal anti-CaMKI antibody (1:500, Sigma, WH0008536M1), and a polyclonal anti–phospho-CaMKI antibody (1:1000, Abcam, ab62215).

**Primary hippocampal neuron cultures and immunofluorescence analysis**. All animal experiments were performed according to the guidelines of the Institute for Developmental Research and were approved by The Animal Care and Use Committee of Nagoya University. Pregnant ICR mice were purchased from SLC Japan. Hippocampal neurons were prepared from E15 mouse embryos with papain as previously described[7, 62]. Briefly, neurons were seeded on coverslips or dishes coated with poly-D-lysine (Sigma) and cultured in neurobasal medium (Invitrogen) supplemented with B-27 (Invitrogen) and 1 mM GlutaMAX (Invitrogen) to observe morphology and perform immunoblot analysis. Neurons were subjected to local application of various factors through micropipettes. Neurons were transfected using Lipofectamine 2000 or Lipofectamine 3000 according to the manufacturer's instructions. Neurons were treated with Y27632 (20 µM), RhoA inhibitor I (4 µg/ml, cytoskeleton) or blebbistatin (50 µM) for 48 h. Neurons were fixed with 4% paraformaldehyde in PBS for 20 min at room temperature and permeabilized in PBS containing 0.1% Triton X-100 and 5% normal goat serum for 20 min. The cells were probed with the indicated primary antibodies followed by secondary antibodies conjugated to Alexa-488, Alexa-546, or Alexa-647 (Invitrogen) as previously described[62–64]. Fluorescence images were acquired with LSM 780 or LSM5 Pascal microscopes constructed around an Axio Observer Z1 or Axiovert 200 M with Plan-Apochromat 20 × (numerical aperture [NA] 0.75), Plan-Apochromat 20 × (NA 0.8), C-Apochromat 40 × (NA 1.2), or Plan Apochromat 63 × (NA 1.40) lenses under the control of LSM software (Carl Zeiss). A neuron was considered to have an axon if one process was at least twice as long as any other process, and a neuron was considered Tau-1-positive, as previously described[7].

**Local application**. Local application of NT-3 (50 µg/ml in a micropipette, PeproTech), anti-NT-3 (500 µg/ml), ionomycin (1 mM), xestospongin C (2 mM), dantrolene (20 mM), SKF96365 (3 mM), ryanodine (100 mM), BAPTA-AM (5 mM), KN-93 (1 mM), Y27632 (20 mM), RhoA inhibitor I (4 mg/ml, Cytoskeleton), or Blebbistatin (50 mM) was performed as previouslydescribed[12, 65] with some modifications. Briefly, an axon or a selected minor neurite was exposed to the factors for 45 or 60 min. Differential interference contrast images were acquired every 5 min with an UplanApo 20 × 1.6 (numerical aperture (NA) 0.75) objective (Olympus) on an inverted microscope (IX81-CSU, Olympus) equipped with an Evolve 512 EMCCD camera (Photometrics) controlled by MetaMorph software (Molecular Devices). In some experiments, the following reagents were applied to the culture medium at least 30 min before the application of the NT-3 gradients as previouslydescribed[12]: 2 µM xestospongin C, 20 µM dantrolene, 3 µM SKF96365, 100 µM ryanodine (Alomone Labs), 5 µM BAPTA-AM (Invitrogen), 1 µM KN-93 or 5 µM STO-609.

**Ca$^{2+}$ imaging**. Hippocampal neurons were loaded with Cal-520TM (2 µM, AAT Bioquest) and F-127 (0.04%, Invitrogen) for 30 min before local application. The images were excited at a wavelength of 492 nm and acquired every 2 s with an UplanApo 20 × 1.6 (numerical aperture (NA) 0.75) objective (Olympus) on an inverted microscope (IX81-CSU, Olympus) equipped with an Evolve 512 EMCCD camera (Photometrics) controlled by MetaMorph software (Molecular Devices). Next, we calculated the imaging ratio for the identified cell region of interest in the field of view. The Ca$^{2+}$ concentration ($R_{treat}/R_0$) was normalized against the baseline pre-stimulation value. The amplitudes of the NT-3-induced increases in the Ca$^{2+}$ concentration were defined as the mean of $R_{treat}/R_0$ during the period from 50 to 80 s after local application.

**Fluorescence-resonance energy transfer**. FRET images of neurons expressing Raichu-RhoA-Clover-mRuby2 (FRET probe) were acquired every 20 s with an UplanApo 20 × 1.6 (numerical aperture (NA) 0.75) objective (Olympus) on an inverted microscope (IX81-CSU, Olympus) equipped with an Evolve 512 EMCCD camera (Photometrics) controlled by MetaMorph software (Molecular Devices). FRET and donor emission images were acquired using the following filters (ex, excitation; em, emission): ex 485/30 nm and em 530/40 nm for Clover and ex 485/30 nm and em 595/70 nm for Clover-mRuby2 FRET. The ratio of mRuby2

to Clover, as determined by MetaMorph, represented the FRET efficiency, which represented the ratio of the RhoA activity. The amplitudes of the NT-3-induced increases in FRET efficiency were defined as the mean during the period from 3 to 15 min after local application of NT-3.

**LOVTRAP-Rho-kinase**. LOV-based photoactivation analysis was performed as previously described with some modification[30]. Briefly, hippocampal neurons were seeded on glass-bottom dishes and then transfected with pCAG-myc-NTOM20-LOV2 and pCAG-myc-mCherry-Zdk1, pCAG-myc-mCherry-Zdk1-RhoA Q63L, pCAG-myc-mCherry-Rho-kinase CAT-Zdk1, or pCAG-myc-mCherry-Rho-kinase CAT KD-Zdk1 at 2 to 3 DIV using Lipofectamine 2000 (Invitrogen) according to the manufacturer's protocols. Confocal images were recorded for 80 min with an LSM780 built around an Axio Observer Z1 with a Plan Apochromat 63 × oil (NA 1.40) (Carl Zeiss) lens. The cell bodies of mCherry-positive neurons were illuminated with a 20-um square of blue light from a 488-nm laser for 5 s between each pair of image acquisitions.

**Mathematical model for the diffusion of Rho-kinase along the neurites**. The dynamics of the spatial distribution of Rho-kinase along a neurite were described as follows:

$$\frac{\partial C}{\partial t} = D\frac{\partial^2 C}{\partial x^2} - kC, \tag{3}$$

with boundary conditions of $C(x=0, t) = C_s(t)$ and $\partial C/\partial x|_{x=L} = 0$, where $t$, $x$, $L$, $C$, $C_s$, $D$ and $k$ indicate time, the space coordinate along a neurite, the length of the neurite, the concentration of Rho-kinase along the neurite, the concentration of Rho-kinase in the cell body, the diffusion rate, and the inactivation and degradation rates, respectively. The time-dependent concentration of Rho-kinase in the soma is represented by $C_s(t) = C_o P(t)$, where $C_o$ and $P(t)$ indicate the Rho-kinase concentrations in the cell body during photoactivation and the time course of photoactivation, in which the binary variables 0 and 1 represent without and with photoactivation, respectively.

Notice that this model did not include active transport of Rho-kinase from the cell body to neurite tips, in contrast to previous models[31, 33, 34, 66]. According to our previous theoretical work[31], actively transported molecules are more accumulated in longer neurites. If Rho-kinase is so, growth of longer neurites is more inhibited by active Rho-kinase. This is inconsistent with our experimental finding that growth of shorter neurites is more inhibited by photoactivation of Rho-kinase (Fig. 3g, h). Moreover, in our imaging, we did not observe active transport of Rho-kinase. Thus, we did not take account for active transport of Rho-kinase.

The steady state distribution of Rho-kinase along a neurite was described by $\partial C/\partial x^2 = (k/D)C$, which was calculated as follows: $C(x) = A\exp(-(k/D)^{1/2}x) + B\exp((k/D)^{1/2}x)$, where $A$ and $B$ represent constants. The general solution satisfying the boundary conditions was as follows:

$$C(x) = \frac{C_0}{1 + \exp(-2\sqrt{k/DL})}\left[\exp(-\sqrt{k/Dx}) + \exp(-2\sqrt{k/DL})\exp(\sqrt{k/Dx})\right], \tag{4}$$

Next, we calculated the amount of Rho-kinase at the neurite tip according to $C(L)$, which is described in equation (1) in the text.

**Model parameters**. The initial neurite length was set as $(F_o + cF_i)/\mu$, where $c$ is a parameter ($0 \le c \le 1$). The parameter values used were $D = 20$ ($\mu m^2\,s^{-1}$), $k = 0.02$ ($s^{-1}$), $\mu = 0.03$ ($min^{-1}$), $F_o = 0.01(\mu\,ms^{-1})$, $K/C_o = 0.4$, $h = 2$ and $c = 0.95$ (Fig. 4b, c, e, c). It has been reported that the diffusion rate of protein in mammalian cell cytoplasm is 20~100 ($\mu m^2\,s^{-1}$)[67], and therefore, we adopted $D = 20$ ($\mu m^2\,s^{-1}$). In response to photoactivation, Rho-kinase is rapidly released within 1 min, which means that its time constant is approximately 60 s, (i.e., $k = 1/60 \fallingdotseq 0.02\,s^{-1}$). The value of $\mu$ was determined by the retraction speed of the neurite during photo-activation (Fig. 3g, h). Values of other parameters ($F_o$, $K/C_o$, $h$, and $c$) were heuristically selected but estimated to fit the experimental data (Fig. 4h); estimated values were $F_o = 0.0134$ ($\mu m\,s^{-1}$), $K/C_o = 0.2756$, $h = 2.7888$ and $c = 0.95$.

**Bottom-up derivation of equation 2**. The neurite growth was modeled by

$$\frac{dL_i}{dt} = v_i, \tag{5}$$

$$m\frac{dv_i}{dt} = -\lambda v_i - \kappa L_i + f_0 + f_i G(R_i), \tag{6}$$

where $L_i$ and $v_i$ indicate the length and growth speed of the $i$th neurite, respectively; $m$, $\lambda$, and $\kappa$ indicate the mass of the growth cone, the viscosity coefficient and the spring constant for the retraction force, respectively; $f_o$, $f_i$, and $R_i$ indicate the constitutive growth force, the Rho-kinase-regulated growth force of the $i$th neurite

and the Rho-kinase concentration at the tip of the $i$th neurite as given by equation (1), respectively. If $m \ll \lambda$, $v_i$ rapidly reaches quasi-steady state,

$$v_i = \frac{-\kappa L_i + f_o + f_i G(R_i)}{\lambda}. \tag{7}$$

Then, equation becomes

$$\frac{dL_i}{dt} = -\frac{\kappa}{\lambda} L_i + \frac{f_o}{\lambda} + \frac{f_i}{\lambda} G(R_i). \tag{8}$$

This equation is namely equation (2), where $\kappa/\lambda$, $f_o/\lambda$, and $f_i/\lambda$ were rewritten as $\mu$, $F_o$, and $F_i$, respectively.

**Reaction-diffusion simulation with a growing neurite.** To validate our approximation of $R_i$ in equation (2), we simulated reaction-diffusion of the Rho-kinase with a moving boundary caused by neurite growth, i.e., $\partial C/\partial x|_{x=L(t)} = 0$. Suppose a spatial scaling of $y = x/L(t)$, where $x \in [0, L(t)]$ and $y \in [0, 1]$. Then, equation (2) was transformed to

$$\frac{\partial C}{\partial t} = \frac{D}{L(t)^2} \frac{\partial^2 C}{\partial y^2} - kC \tag{9}$$

with boundary conditions of $C(y = 0, t) = C_s(t)$ and $\partial C/\partial y|_{y=1} = 0$. Based on equation (9), we performed an exact simulation of the reaction-diffusion process coupled with neurite growth. We then confirmed that the steady state approximation was valid (Supplementary Fig. 5d–f).

**Non-linear regression method.** We performed a regression in which the parameters in equations (1 and 2), denoted in total by $\theta$, were estimated by non-linear regression to minimize the quadratic error function as follows:

$$E(\theta) = \frac{1}{2} \sum_{n=1}^{M} \{t_n - f(z_n; \theta)\}^2, \tag{10}$$

Where $f(z_n; \theta)$ is the simulated retraction length depending on the initial length of the neurite, as shown in Fig. 4f, which we aimed to fit to the experimental data; $z_n$ and $t_n$ are the observed length (i.e., minor neurites and axon) at the onset of photoactivation and the observed retraction length after photoactivation, respectively, of the $n$th neurite; and $M$ indicates the total number of neurites among all neurons. We used a simplex Nelder–Mead algorithm to minimize the error function (10).

**Mass spectrometry (KISS method).** Rat brain lysate was incubated with glutathione Sepharose beads coated with GST-CaMKI-cat (500 pmol). The proteins bound to CaMKI-cat were incubated with or without ATP and $Mg^{2+}$ and then denatured with guanidine hydrochloride and digested with trypsin for 16 h at 37 °C following reduction, alkylation, demineralization, and concentration. The phosphopeptides were concentrated using a Titansphere Phos-TiO Kit (GL Sciences) according to the manufacturer's instructions. Nanoelectrospray tandem mass analysis was performed using a Q Exactive mass spectrometry system (Thermo Fisher Scientific Inc.) combined with an HTC-PAL autosampler and a Paradigm MS4 HPLC System (Michrom BioResources Inc.) using a C18 reverse-phase column and a Michrom ADVANCE Plug-and-Play NanoSpray Source. A peak list was generated and calibrated using MaxQuant software. Database searches were performed against the complete proteome set of *Rattus norvegicus* in UniProt KB 2014_07 concatenated with reverse copies of all sequences. The ion intensities of the identified peptides in the phosphorylated sample were compared with those of the nonphosphorylated sample, and the phosphorylation sites exhibiting greater than fivefold increases in ion intensity were regarded as candidate substrates[36].

**In vitro phosphorylation assay.** The phosphorylation assay was performed as previously described[68]. Briefly, kinase reactions for CaMKI were performed in 100 µl of a reaction mixture (50 mM Tris/HCl, pH 7.5, 1 mM EDTA, 1 mM EGTA, 1 mM DTT, 5 mM $MgCl_2$, and 100 µM [$\gamma$-$^{32}$P] ATP [1-20 GBq/mmol]), with 0.05 µM of the catalytic domain of CaMKI and purified GST-GEF-H1 mutants for 30 min at 30 °C. The reaction mixtures were then boiled in sodium dodecyl sulfate (SDS) sample buffer and subjected to SDS-polyacrylamide gel electrophoresis (PAGE) and silver staining. The radiolabeled proteins were visualized with an image analyzer (BAS1500; Fuji). Full-length blots are shown in Supplementary Fig. 10.

**Analysis of GEF activity of GEF-H1.** The guanine nucleotide exchange activity of GEF-H1 was measured as previously described with some modifications[40, 61]. In brief, GST-RhoA-G17A was transfected into COS-7 cells (obtained from RIKEN Cell Bank, RCB0539). The cells were lysed in 10 mM HEPES, pH 7.5; 150 mM NaCl; 1 mM $MgCl_2$; 1 mM EGTA; 0.2% Triton-X; protease inhibitor mixture

(cOmplete Mini EDTA-free, Roche); and phosphatase inhibitor mixture (PhosSTOP, Roche). The cell lysate, which was obtained by centrifugation at 15,000×$g$ for 10 min at 4 °C, was incubated with glutathione Sepharose beads (GE Healthcare). After centrifugation at 2500×$g$ for 5 min, the GST-RhoA-G17A-glutathione Sepharose beads were incubated at 4 °C in extracts of COS-7 cells expressing GEF-H1. SDS-PAGE and immunoblotting were performed as previously described[7, 63]. GEF-H1 band intensities were quantified using ImageJ software. Full-length blots are shown in Supplementary Fig. 10.

**In utero electroporation.** In utero electroporation was performed as previously described[12, 27, 41] with some modifications. pTα-1-LPL-Lyn-EGFP (2 µg/µl) was co-microinjected with pTα-1-Cre (0.01 µg/µl) and pTα-LPL-GEF-H1–WT, –T103A or –T103E (4 µg/µl) into the lateral ventricle of E13 embryos. To achieve GEF-H1 knockdown, pTα-LPL-Lyn-EGFP was coinjected with pSico-mCherry-shGEF-H1#1 or - shGEF-H1#1 (3 µg/µl), and pTα-Cre (0.05 µg/µl). After plasmid injection into the lateral ventricle of the embryos at E13, electric pulses (50-ms square pulses of 32 or 40 V with 950-ms intervals) were applied to the embryos.

**Immunohistochemistry and quantitative analysis.** Immunohistochemistry was performed as previously described[12, 27, 41]. Briefly, brains were fixed in 4% paraformaldehyde at E16.5 and coronally sectioned with a cryostat (Leica Microsystems) at a thickness of 60 µm. The slices were incubated with primary antibodies diluted in PBS containing 1% bovine serum albumin (BSA) and 0.01% Triton X-100 at 4 °C overnight followed by Alexa Fluor 488- or Alexa Fluor 555-conjugated secondary antibodies diluted in PBS containing 1% BSA and 0.01% Triton X-100 for 1 h at room temperature. The nuclei were visualized by staining with Hoechst 33342 (Invitrogen). Confocal images were recorded using an LSM 780 built around an Axio Observer Z1 with Plan-Apochromat 20 × (numerical aperture (NA) 0.75), C-Apochromat 40 × (NA 1.2), or Plan Apochromat 63 × (NA 1.40) lenses under the control of LSM software (Carl Zeiss). Coronal sections of cerebral cortices containing the labeled cells were classified into CP and IZ, which were outer layer and middle layer, respectively. The number of all labeled cells in each region was calculated. To evaluate the morphology of the migrating neurons, projection images of EGFP-positive neurons were obtained from Z-series confocal images using LSM software. At least three independent fetal brains were electroporated and analyzed for each experiment. All experiments were performed in a blinded manner.

**Statistical analysis.** Data are expressed as the mean ± SEM. The sample size was defined based on our previous reports[12, 41] and G*Power 3.1 software. All statistical analyses were performed with GraphPad Prism version 6 (GraphPad Software) using Student's $t$-tests (two-tailed), Dunnett's tests, or Tukey's multiple-comparison tests. $P < 0.01$ and $P < 0.05$ were considered to indicate statistical significance, and the effective size was calculated based on a previous report[69]. All results of the statistical analysis are shown in Supplementary Table 2.

**Data availability.** The data that support the findings of this study are available from the corresponding author on request.

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

## Acknowledgements

We thank Dr H. Miki for the plasmids, The Division for Medical Research Engineering (I. Mizuguchi and Y. Itoh) at Nagoya University Graduate School of Medicine for technical assistance, and F. Ishidate for help with image acquisition. We also thank T. Ishii for secretarial and technical assistance. This work was supported by the Japan Science and Technology Agency, Core Research for Evolutionary Science and Technology, a Grant-in-Aid for Scientific Research on Innovative Areas (25123705) and a Grant-in-Aid for Scientific Research (25251021), a Grant-in-Aid for Young Scientists (B) (26830045), a Grant-in-Aid for JSPS Fellows (PD) (20153173) from the Japan Society for the Promotion of Science and a Grant-in-Aid for Bioinformatics for brain sciences conducted under the Strategic Research Program for Brain Sciences and the Platform for Dynamic Approaches to Living Systems from the Ministry of Education, Culture, Sports, Science (MEXT) and Japan Agency for Medical Research and development (AMED), and by the National Institutes of Health grants GM103723 and EB002025.

## Author contributions

K.K. and T.T. designed the study. T.T., M.W., and N.I. performed the time-lapse imaging. M.A., Y.Y., and T.H. performed the phospho-proteomic analysis. S.N. produced the antibodies. T.W. and K.H. produced the LOVTRAP-Rho-kinase. N.H. performed the mathematical modeling. T.T., M.W., and N.I. performed the biological experiments and the morphological analysis of the neurons. T.T., M.W., N.I., C.X., and T.N. performed the immunohistochemistry. All authors discussed the results and commented on the manuscript text.

## Additional information

**Competing interests:** The authors declare no competing financial interests.

