## [Peer Review File · Nature Communications]

Reviewers' comments:

Reviewer #1 (Remarks to the Author):

The manuscript by Takano et al. reports that local application of neurotrophins in growing axons acts to suppress the dendritic growth via a long distance calcium-CaMKI-GEF-H1-RhoA pathway. Most previous studies of neuronal polarity focused on axon-dendrite specification. The results in this study aimed to provide a mechanistic explanation of how hippocampal neurons can maintain a single axon after establishing initial neuronal polarity. The study was well performed and many advanced live cell imaging techniques were used. The roles of RhoA and CaMKI in neuronal polarization are well known. But the findings of how RhoA and CaMKI are spatially regulated by calcium wave and the identification of GEF-H1 are novel. There are several major concerns that need to be addressed.

1. The study was mainly focusing on the role of local NT-3 signaling in maintaining a single axon after establishing neuronal polarity. The authors should make this very clear throughout the manuscript.

2. Another major concern is that the study was performed with local application of NT-3. As recognized by the authors in the discussion, hippocampal neurons under control condition without NT-3 treatment also grow a single axon. The authors stated that under control condition, local neurotrophin production (autocrine) might act similarly. If this is the case, can inhibition of neurotrophin autocrine or removal of self-produced neurotrophin (e.g. with function blocking antibody) affect polarized axon growth of control neurons? In addition, can abolishing calcium wave (e.g. with drugs used in Figure 2b) affect polarized axon growth of control neurons? Furthermore, in Figure 2c, PBS application did not induce CaMKI phosphorylation in the cell body. Does this result suggest that CaMKI activation is not required for polarized axon growth of control neurons? The answers to these question are very important to show convincingly that the identified mechanism is not just specific for local NT-3 induced effect.

3. In Supplementary Figure 1, the authors showed that local application of ionomycin to the cell body induced minor neurite retraction. Will the same treatment in the axon also induce axon retraction? NT-3 induced calcium increased could be different from ionomycin-induced calcium increase.

4. In Figure 3a, it seems that NT-3 application also activated Rho in distal axons compared with PBS gradient. In 3i, it showed that axon growth was not affected by local activation of Rho in the cell body, right? Will local activation of Rho in the axon induce axon retraction? More important to the pathway proposed by the authors, will local inhibition of Rho/ROCK/myosin II antagonize NT-3 gradient-induced dendrite retraction?

5. In Figure 7, the effects of GEF-H1 manipulation on polarized axon growth were not very obvious compared with Rho/ROCK inhibition shown in Figure 5. At least in 7g, GEF-H1T103A induced Tau1 positive neurites are much shorter than the original axon. The authors should quantify the length ratios among Tau1 positive neurites.

6. The in vivo results shown in Supplemental Figure 8 is actually very important and should be moved to the main figure and better explained.

In summary, the manuscript includes a lot of experimental results. To my opinion, it will greatly strengthen the manuscript if the authors could present their results in a way emphasizing their main conclusion and novel findings.

Reviewer #2 (Remarks to the Author):

This is a comprehensive paper on an important neurodevelopmental question, namely how do neurons develop polarity. The authors define a novel signaling pathway in hippocampal neurons that ensures the development of a single long axon and multiple minor neurites that develop into dendrites. According to the stochastic model of neuronal polarization axonal fate is stochastically determined in the absence of additional extracellular factors. However, neurotrophins are essential for cultured hippocampal neuronal polarization even in the stochastic model. Here the authors investigate the long range inhibitory signaling that permits polarized neurons to develop only a single axon by preventing the minor neurites from forming unnecessary axons. The paper is well written and the results are novel and convincing, employing a battery of cutting edge techniques to define an inhibitory pathway that originates from NT3 induced calcium waves in the axon that travel back to the cell body and can also invade the minor neurites. These waves in turn increase RhoA activity in the cell body through CAMKI. Rho kinase diffuses through the minor neurites but not the axon. This inhibits the growth of minor neurites by a signaling pathway involving the phosphorylation of a RhoA specific GEF, GEF-HI, whose phosphorylation enhances its GEF activity. In general the data are compelling and well support the conclusions of the paper. However, I have the following comments and suggestions.

1. Figure 1 nicely shows that local application of NT-3 and BDNF to the axon terminal elicits axon outgrowth but shortens the length of minor neurites.

2. Figure 2 shows that local NT-3 application evokes transient intracellular calcium waves that travel back to the cell body and sometimes invade minor neurites. When inhibitors are applied calcium elevation is eliminated. However my repeated viewing of the accompanying video seems to show elevated calcium activity in the cell body and minor neurites occurring simultaneously or prior to the calcium wave traveling retrogradely to reach the cell body. Can the authors comment on this? Next the authors investigate a target of calcium activity, CAMKI by applying NT-3 and staining for phosphorylated CAMKI, which is increased in the cell body and axon concomitant with retraction of minor neurites. Application of calcium and CAMKI inhibitors nicely shows that retraction of minor neurites is dependent on this calcium signaling.

3. Figure 3 demonstrates with FRET that activation of RhoA in the cell body and in the minor neurites occurs downstream of calcium signaling induced by NT-3. (A minor point: the videos referred to in the text as videos 3-4 are actually videos 4-5 in the supplemental data

and in the legends that accompany them.) Application of inhibitors clearly demonstrates that RhoA activation is mediated by long range calcium signaling. As further support for the role of Rho-kinase in minor neurite retraction, the authors use an optogenetic approach to activate Rho-kinase in the cell body which results in retraction of minor neurites. In the middle (experimental) panel of Fig 3e some but not all of the minor neurites show visible shortening even though cumulatively minor neurites retract. What accounts for this? Since minor neurites actually extend and retract in the videos during the 45 min observation period it was difficult to evaluate actual retraction in comparing the videos in experimental and kinase dead conditions. Can the authors state whether the minor neurites that shorten also show activation of RhoA with FRET as shown in supplementary Figure 2? This is important since Rho-A activation is the stated mechanism for minor neurite contraction.

4. To test diffusion of activated Rho-kinase into short minor neurites vs. the axon, the authors develop a mathematical model to show that Rho-kinase accumulates to a greater degree in minor neurites than in axons. This is shown by computer simulations, which I am unable to evaluate. The model for neurite retraction suggests that Rho-kinase regulated growth force is length-dependent so Rho-kinase would affect short but not long neurites, consistent with its ability to diffuse into short minor neurites but not axons.

5. In Figure 5 the authors ask whether prevention of outgrowth of minor neurites by active Rho-kinase is the mechanism for ensuring formation of a single long axon. Using gradients of a Rho-kinase inhibitor, the authors show with live cell imaging that this promotes rapid growth of a minor neurite. The video nicely shows this and reveals that the typical extension and retraction behavior does not occur. Can the authors comment on whether this steady forward extension is typical of minor neurites treated with the Rho-kinase inhibitor? This figure shows that application of Rho-kinase inhibitors to hippocampal neurons increases the numbers of neurons with multiple axons. Interestingly as shown in supplemental Fig 4, newly formed tau labeled axons grow from the tips of minor neurites identified with MAP2. Since the two markers overlap it raises the question of how the new process is converted from a minor neurite to an axon.

6. In Fig 6 the authors investigate interaction between CAMKI and RhoA/Rho-kinase. Using a novel screening technique developed by their laboratory, the authors identify a number of candidate phospho proteins as substrates for CAMKI but focus on GEF-H I as the only specific RhoA regulator. The authors identify the Thr103 site on GEF-HI directly phosphorylated by CaMKI. Under physiological conditions when hippocampal neurons are stimulated with NT-3 GEF-H1 phosphorylation at Thr103 is increased and this increase in the cell body is clearly demonstrated in Supplemental figure 6.

7. As shown in Fig 7 the GEF-HI is required for the activation of RhoA, which underlies the retraction of minor neurites, and the computer model shows diffusion of activated Rho-kinase in minor neurites so I wonder whether the large increase in GEF-HI phosphorylation in the cell body also occurs in the minor neurites. Do the authors ever observe this? GEF-HI knockdown (supplemental Fig 7) or transfection of hippocampal neurons with a non phosphorylatable mutation of GEF-HI at T103 (Fig 7g) caused formation of multiple axons. Would the elimination of the restraint on minor neurites and the resulting growth of multiple

axons by GEF-HI inactivation require its expression in minor neurites?

8. In the final figure 8 (Supplemental) the authors test the relevance of this RhoA signaling pathway to the development of neuronal polarization and migration of cortical pyramidal neurons in vivo. Using in utero electroporation of GEF-HI to knock down or express WT, phosphomimetic (T103E) and non-phospho (T103A) constructs, the authors found that overexpression of WT GEF-HI or the T103 and T103A mutants all caused failure of migration into the cortical plate with most neurons lacking a leading process in contrast to control bipolar neurons. However, the morphology of the T103E vs T103A expressing neurons had very different morphologies i.e. round shapes with few processes vs. long trailing processes. How do the authors interpret these results in light of their in vitro findings? GEF-HI knockdown also reduced pyramidal neuron migration into the cortical plate and fewer neurons had bipolar morphologies. These defects were rescued with GEF-HI WT electroporation. In my view these in vivo experiments are important and should be presented in the body of the paper with a fuller discussion rather than as a supplemental figure.

9. This interesting paper uses a variety of techniques to present a novel and mechanistic model of how neurotrophins, by activating a retrograde inhibitory calcium signaling pathway involving RhoA contractility of minor neurites, can induce polarity in hippocampal neurons resulting in a single long axon and multiple short dendrites. Given the extensive and convincing data in the paper, I found the Discussion somewhat flat in that the authors mostly summarized the results. For example the authors raise an important question regarding development of neuronal polarity in the absence of exogenous neurotrophins (i.e. a stochastic model) but do not fully address this issue. Would similar signaling pathways occur in the absence of neurotrophins? It would also be interesting for the authors to discuss how their intriguing in vivo results on cortical migration and neuronal morphology relate to their findings in cultured hippocampal neurons, which was not clear from their description of supplemental figure 8. Nevertheless this paper should be of wide interest to researchers interested in the important mechanisms of neuronal polarity.

Reviewer #3 (Remarks to the Author):

Taken et al. propose that the Ca²⁺-CaMKI-GEF-H1-RhoA-ROCK signaling axis is involved in the long range inhibitory signaling to maintain neuronal axon-dendrite polarity. The authors showed a number of interesting findings: 1) NT-3 and BDNF elongate only axons, but not dendrite; 2) The treatment of these neurotrophic factors elevated Ca²⁺ in axon shaft, which is transmitted to cell body and activates downstream CaMKI; 3) CaMKI phosphorylates GEF-H1 and activates RhoA and ROCK/Rho kinase, one of the downstream factors of RhoA, which is involved in neurite retraction; 4) Optogenetical activation of ROCK confirmed that ROCK activation in minor neurites is sufficient to suppress outgrowth. Overall findings are interesting and results are of high quality. However, this study provides only in vitro culture results. The biological significance of these very interesting findings are not validated in vivo. There are also a few missing experiments, which would make the story more convincing

and complete.

1) In Figure 1, the authors locally applied two neurotrophins gradient, NT-3 and BDNF, to enhance the inhibitory signals from axon. However, no attempt was made to test the in vivo relevance about the neurotrophin gradient of these neurotrophins. Do these factors distributed in a gradient in hippocampus and developing cortex in vivo?

2) In Figure 2, calcium wave was observed after local application of NT-3 to single axon. The treatment of four different inhibitors suggests that ryanodine receptor is involved in Ca-wave transition through axon to cell body/dendrite. However, the experiments that test how these inhibitors affect axon and minor neurites outgrowth are missing.

3) In Figure 3, the authors used Raichu-RhoA probe to monitor RhoA signaling activity. The authors mentioned "RhoA activity" in the main text (page 8, line 136), however, this probe can detect the activity of GEF against RhoA (more precisely, the balance of activity between GEF and GAP), not directly detect RhoA activity. This needs to be clearly described.

4) In Figure 4, the authors used computational simulation to monitor ROCK distribution after photoactivation (in Figure 3). Simulation appears to be excellent way to monitor ROCK distribution, but in this case, ROCK-Zdk is overexpressed in the neurons, which unlikely represent endogenous ROCK localization. It would be better to simulate endogenous ROCK localization under control (stochastic model) and NT-3-treated condition, if possible. Is there any good antibody to detect endogenous ROCK? Moreover, the authors should demonstrate the level of ROCK overexpression in comparison to endogenous ROCK.

5) In Figure 7, the authors showed that WT GEF-H1 can be localized in not only cell body/dendrite, but also axon shaft. This raises another questions why RhoA-ROCK-shrinkage signaling is not activated in an axon, because Ca and CaMKI are also activated in axons after local application of NT-3 to axons. This needs to be addressed.

6) To assess the GEF-H1 effect on axon-dendrite polarity in vivo, the authors performed in utero electroporation into telencephalon. This is an excellent experiment, however, the authors should also evaluate CaMKI, RhoA and ROCK function in these experimentsn to test the role of this signaling axis.

Reviewer #4 (Remarks to the Author):

Takano et al. propose a long-range signalling mechanism involved in the suppression of the growth of minor neurites. As asked by the editor, the below comments are focussed on the modelling aspects of the manuscript.

The authors construct a mathematical model to mimic their optogenetic LOVTRAP experiments. First, using reaction-diffusion formalism, they describe the spatial distribution of active Rho-kinase in neurites during optogenetic activation. Second, using an ordinary differential equation model of neurite growth, they describe how activation levels of Rho-

kinase at the tips of neurites affect its growth. The model predicts that activated Rho-kinase diffuses from the cell body to the tips of small minor neurites but not to the tips of long axons. There are a number of critical issues with the model formulation and assumptions (see below). Therefore, the biological insights gained from the model predictions are unclear.

Specific comments:

1. In Eq. (1), should it be $(\cosh(L/\sqrt{D/k}))^{-1}$ (which is $\operatorname{sech}(L/\sqrt{D/k})$) instead of $\cosh^{-1}(L/\sqrt{D/k})$ (which is $\operatorname{arcosh}(L/\sqrt{D/k})$)?
2. The authors assume that the reaction-diffusion process attains steady state rapidly compared to the time scale of optogenetic experiments (~ 60 minutes). Based on $\sqrt{D/k} = 31.6 \mu\text{m}$ (given in figure 4 legend), the half-life of active Rho-kinase $(\ln(2)/k)$ works out to be ~ 11.5 minutes (for $D = 1 \mu\text{m}^2 \text{ s}^{-1}$) and ~ 115 minutes (for $D = 0.1 \mu\text{m}^2 \text{ s}^{-1}$). The half-life appears to be comparable to the time scale of experiments. Thus, it is not clear if the system attains steady state rapidly. The authors should describe the parameter regime (D and k) in which their steady state assumption holds and discuss if the parameter regime is biologically reasonable.
3. Previous mathematical models of axon-dendrite specification consider both diffusion and active transport (anterograde and retrograde) of molecules between the soma and the neurite tips (for e.g., refs. [1-3]). Active transport could enrich molecules at the axon tips. Is there a reason to neglect transport of active RhoA kinase?
4. The neurite growth model (Eq. (2)) predicts that, during photoactivation of Rho-kinase, the neurite length does not change for large values of initial neurite length (Figs. 4e and 4h). This prediction appears to be inconsistent with the experimental data where the axon length appears to increase during photoactivation (Fig. 3g). Shouldn't axons grow at the normal rate when the photoactivated Rho-kinase level is very low at the axon tips?
5. The rate $-\mu L_i$ in the Eq. (2) indicates that the neurite retraction rate increases with neurite length. I find it hard to understand why retraction rate should increase with neurite length. Is there any direct experimental evidence or biophysical reasoning for this assumption?
6. The parameter μ in the Eq. (2) is defined as Young's modulus for retraction force. The unit of Young's modulus is the pascal, whereas the unit of μ in Eq. (2) is s^{-1} . It is unclear what μ means in the equation.
7. As per the Eq. (2), the maximum length that any neurite can grow is $(F_0 + F_i)/\mu$ (assuming $R_i = 0$). It is not clear if such a limit exists for neurite growth, as axons can grow until synaptic connections are established.
8. The authors estimate six model parameters (μ , F_0 , F_i , $\sqrt{D/k}$, h and K/C_0) from one data set (Fig. 4h). The authors must demonstrate that they are not overfitting the data. Perhaps the authors could use the control experiments (Fig. 3f) to constrain some of the

model parameters such as μ and F_0 .

9. The authors use $P = 0.1$ as a threshold for statistical significance in Fig. 1 and Supplementary Fig. 1, and $P = 0.05$ elsewhere. The authors will have to justify this in the manuscript.

10. How did the authors determine the sample size of their experiments? The authors must discuss in the manuscript whether the chosen sample size is sufficient to measure their effect size (see ref. [4]).

11. It is not clear if the n values in the figure legends indicate number of cells or number of independent experiments. Please mention both in the figure legends.

Minor points:

1. Page 44, line 789, equation number should be (5) instead of (4).
2. Please provide all the estimated parameter values (for instance, F_i value is missing).

References:

1. Toriyama M, Sakumura Y, Shimada T, Ishii S, & Inagaki N (2010) A diffusion-based neurite length-sensing mechanism involved in neuronal symmetry breaking. *Mol Syst Biol* 6:394.
2. Samuels DC, Hentschel HGE, & Fine A (1996) The origin of neuronal polarization: A model of axon formation. *Philos T Roy Soc B* 351(1344):1147-1156.
3. Fivaz M, Bandara S, Inoue T, & Meyer T (2008) Robust neuronal symmetry breaking by Ras-triggered local positive feedback. *Current Biology* 18(1):44-50.
4. Sullivan GM, Feinn R (2012) Use effect size-or why the P value is not enough. *J Grad Med Educ* 4(3):279-282.

Our responses to the reviewers' comments and changes in the revised version

Thank you very much for reviewing our manuscript. We appreciate your efforts toward understanding our work and your valuable comments on the manuscript. We have attempted to address all the questions and concerns. We believe that additional data have substantially strengthened our conclusions and greatly emphasized the novelty of our findings.

Reviewer #1:

Thank you for your valuable comments. We have revised the manuscript accordingly.

1) The study was mainly focusing on the role of local NT-3 signaling in maintaining a single axon after establishing neuronal polarity. The authors should make this very clear throughout the manuscript.

According to the reviewer's suggestion, we have clearly described the NT-3-dependent long-range inhibitory signaling as our proposed model throughout the revised manuscript (Page 2, lines 6-8; Page 21, lines 13-15; Page 22, lines 1-2; Page 22, lines 16-18).

2) Another major concern is that the study was performed with local application of NT-3. As recognized by the authors in the discussion, hippocampal neurons under control condition without NT-3 treatment also grow a single axon. The authors stated that under control condition, local neurotrophin production (autocrine) might act similarly. If this is the case, can inhibition of neurotrophin autocrine or removal of self-produced neurotrophin (e.g. with function blocking antibody) affect polarized axon growth of control neurons? In addition, can abolishing calcium wave (e.g. with drugs used in Figure 2b) affect polarized axon growth of control neurons? Furthermore, in Figure 2c, PBS application did not induce CaMKI phosphorylation in the cell body. Does this result suggest that CaMKI activation is

not required for polarized axon growth of control neurons? The answers to these questions are very important to show convince that the identified mechanism is not just specific for local NT-3 induced effect.

We thank the reviewer for these important comments. In previous papers (Nakamuta et al., 2011; Cheng et al., 2011), inhibition of neurotrophin signaling by inhibitors, neutralizing antibodies or dominant negative mutants suppressed axon formation in the stochastic model. Furthermore, we have reported that Trk receptors are accumulated at the distal part of the axon in a Kinesin-1 and CRMP-2-dependent fashion (Arimura et al., 2009). Based on these findings, neurotrophin signaling could be amplified at the nascent axon and, in turn, regulate axon formation and prevent other minor neurites from forming an axon through long-rang inhibitory signaling in the stochastic model. We have described and discussed these previous findings on page 4, lines 1-3 and page 22, lines 7-10 in the revised manuscript.

According to the reviewer's suggestion, we examined whether the attenuation of long-rage Ca^{2+} propagations induced minor neurite elongation using several specific inhibitors. Local application of inhibitors, including ryanodine receptor inhibitors, to an axon induced minor neurite elongation. In contrast, local application of xestospongine C (IP_3 receptor inhibitor) suppressed axonal outgrowth, whereas ryanodine, dantrolene (ryanodine receptor inhibitors) or SKF96365 (a TRP channel inhibitor) did not. These results indicate that long-range inhibitory signaling plays a critical role in neuronal polarization not only in the NT-3-stimulated model but also in the stochastic model. We have described the results on page 8, lines 2-8, and have added these results to Fig. 2g and Fig.2h in the revised manuscript.

As the reviewer pointed out, we have carefully reexamined our present results and found that slight phospho-CaMKI could be observed in the cell body of control neurons (Fig. 2e). In addition, to examine whether CaMKI activity in the cell body of hippocampal neurons inhibits minor neurite outgrowth, we locally exposed the cell body to CaMKI inhibitor (KN-93) as suggested by the reviewer and as described

below (Reviewer #2, comment 2). Local application of KN-93 to the cell body induced minor neurite elongation, indicating that CaMKI activity in the cell body is required for neuronal polarization even in the stochastic model. We have described these results on page 8, lines 8-10, and have added these results to Supplemental Fig. 1d-e in the revised manuscript.

3) In Supplementary Figure 1, the authors showed that local application of ionomycin to the cell body induced minor neurite retraction. Will the same treatment in the axon also induce axon retraction? NT-3 induced calcium increase could be different from ionomycin-induced calcium increase.

According to the reviewer's suggestion, we locally exposed the axon terminals to ionomycin. We found that local application of ionomycin induced minor neurite retraction but not axon retraction, which is consistent with the observations of local application of ionomycin to the cell body. In addition to NT-3-elevated intracellular Ca^{2+} , it has been reported that ionomycin-elevated intracellular Ca^{2+} leads to Ca^{2+} -induced Ca^{2+} release through ryanodine receptors (Randriampita et al., 1991; Kochegarov et al., 2001). Therefore, the ionomycin-induced minor neurite retraction may be mediated by Ca^{2+} -induced Ca^{2+} release from ryanodine receptors, which was similar to the effect of NT-3 stimulation. We have described these results on pages 7-8, and have added these results to Supplemental Fig. 1a-c in the revised manuscript.

4) In Figure 3a, it seems that NT-3 application also activated Rho in distal axons compared with PBS gradient. In 3i, it showed that axon growth was not affected by local activation of Rho in the cell body, right? Will local activation of Rho in the axon induce axon retraction? More important to the pathway proposed by the authors, will local inhibition of Rho/ROCK/myosin II antagonize NT-3 gradient-induced dendrite retraction?

Because RhoA activity in the axon fluctuates during NT-3 stimulation (Supplemental Fig. 2), we have changed the image in Fig. 3a of the revised manuscript. According to the reviewer's suggestion, we first examined whether local activation of RhoA in the cell body could induce minor neurite retraction but not axon retraction using LOVTRAP-RhoA (Wang et al., 2016) in the revised version. We found that the photoactivation of LOVTRAP-RhoA in the cell body induced rapid minor neurite retraction. In contrast, LOVTRAP-RhoA did not affect axon length. Second, we examined the effect of increases in active RhoA/Rho-kinase in the distal part of the axon on axonal outgrowth. We found that photoactivation of RhoA or Rho-kinase but not the kinase-dead form of Rho-kinase in the axon suppressed axon elongation compared with control neurons, indicating that the axon outgrowth is associated with a small amount of active RhoA/Rho-kinase in the distal part of the axon. Third, we examined whether the NT-3-induced RhoA activation was responsible for minor neurite retraction using the following specific inhibitors: C3 (RhoA inhibitor), Y-27632 (Rho-kinase inhibitor) and blebbistatin (myosin II inhibitor). All of the inhibitors restored the minor neurite retraction induced by NT-3 to the levels observed in control neurons. We have described these results on pages 10-11, page 11, lines 3-8, page 9, lines 7-11 and have added these results to Fig. 3a, Fig. 3c and Fig. 3f-i in the revised manuscript.

5) In Figure 7, the effects of GEF-H1 manipulation on polarized axon growth were not very obvious compared with Rho/ROCK inhibition shown in Figure 5. At least in 7g, GEF-H1T103A induced Tau1 positive neurites are much shorter than the original axon. The authors should quantify the length ratios among Tau1 positive neurites.

According to the reviewer's suggestion, we performed a quantification analysis and found that the expression of GEF-H1-T103A significantly increased the ratio of the Tau-1-positive axonal length (1.3 ± 0.06). Treatment with RhoA or Rho-kinase inhibitor also increased the ratio of the Tau-1-positive axonal length (1.66 ± 0.11 and

1.70 ± 0.14, respectively). There was a difference in the experimental time window between Fig. 5c-g and Fig. 7g-k. In Fig. 5c-g, hippocampal neurons were treated with RhoA/Rho-kinase inhibitors at 3 DIV and then observed at 5 DIV. On the other hand, GEF-H1-T103A was transfected into neurons at 3 DIV and then observed at 4 DIV in Fig. 7g-k, and therefore, the increase in multiple GEF-H1-T103A-induced axons was slightly smaller than that of RhoA/Rho-kinase inhibition. We have provided more understandable information of these methods on pages 14-15, page 18, lines 18-19, changed the image of the GEF-H1-T103A-expressing neuron in Fig. 7g and have added these results to Fig. 5e and Fig. 7i in the revised manuscript.

6) The *in vivo* results shown in Supplemental Figure 8 is actually very important and should be moved to the main figure and better explained.

We have added the results regarding *in vivo* neuronal polarization to Fig. 8 in the revised manuscript.

In summary, the manuscript includes a lot of experimental results. To my opinion, it will greatly strengthen the manuscript if the authors could present their results in a way emphasizing their main conclusion and novel findings.

We greatly appreciate your positive comments on our revised manuscript.

Reviewer #2:

We greatly appreciate your positive comments on our manuscript.

1) Figure 1 nicely shows that local application of NT-3 and BDNF to the axon terminal elicits axon outgrowth but shortens the length of minor neurites.

Thank you for your positive comments on our manuscripts.

2) Figure 2 shows that local NT-3 application evokes transient intracellular calcium waves that travel back to the cell body and sometimes invade minor neurites. When inhibitors are applied calcium elevation is eliminated. However my repeated viewing of the accompanying video seems to show elevated calcium activity in the cell body and minor neurites occurring simultaneously or prior to the calcium wave traveling retrogradely to reach the cell body. Can the authors comment on this? Next the authors investigate a target of calcium activity, CAMKI by applying NT-3 and staining for phosphorylated CAMKI, which is increased in the cell body and axon concomitant with retraction of minor neurites. Application of calcium and CAMKI inhibitors nicely shows that retraction of minor neurites is dependent on this calcium signaling.

Because Ca^{2+} propagation from the ER-store oscillates and Ca^{2+} propagation is extremely fast with a velocity of approximately 50 - 100 $\mu\text{m/s}$ (Collier et al., 2000; Nakayama et al., 2002; Ross., 2012; Neymotin et al., 2015), it is practically difficult to visualize the dynamics of a single Ca^{2+} wave within a neuron. Nevertheless, we found that the transient intracellular Ca^{2+} waves gradually increased in the axonal shaft toward the cell body, and subsequently, the concentration of Ca^{2+} in the cell body was elevated after local application of NT-3 (Fig. 2a-b). These results suggest that Ca^{2+} waves are generated by NT-3 and propagate from the axon to the cell body. Characterization of the dynamics of a single Ca^{2+} propagation within in a neuron would be another major project that is beyond the scope of the present study using advanced imaging technology.

We agree with this suggestion. Given that CaMKI was more activated in the cell body of control neurons than in the axon and minor neurites (Fig. 2e), we locally exposed the cell body to BAPTA (Ca^{2+} chelator) and KN-93 (CaMKI inhibitor). Local application of BAPTA or KN-93 to the cell body induced minor neurite elongation. We

have described the results on page 8, lines 8-10, and added these results to Supplemental Fig. 1d-e in the revised manuscript.

3) Figure 3 demonstrates with FRET that activation of RhoA in the cell body and in the minor neurites occurs downstream of calcium signaling induced by NT-3. (A minor point: the videos referred to in the text as videos 3-4 are actually videos 4-5 in the supplemental data and in the legends that accompany them.) Application of inhibitors clearly demonstrates that RhoA activation is mediated by long range calcium signaling. As further support for the role of Rho-kinase in minor neurite retraction, the authors use an optogenetic approach to activate Rho-kinase in the cell body which results in retraction of minor neurites. In the middle (experimental) panel of Fig 3e some but not all of the minor neurites show visible shortening even though cumulatively minor neurites retract. What accounts for this? Since minor neurites actually extend and retract in the videos during the 45 min observation period it was difficult to evaluate actual retraction in comparing the videos in experimental and kinase dead conditions. Can the authors state whether the minor neurites that shorten also show activation of RhoA with FRET as shown in supplementary Figure 2? This is important since Rho-A activation is the stated mechanism for minor neurite contraction.

According to the reviewer's minor point, we have corrected the video numbers. We agreed with the reviewer's comment. Our mathematical model clearly showed a U-shaped relationship between neurite length at the onset of photoactivation and neurite retraction length after photoactivation of Rho-kinase (Fig. 4h). These results suggest that Rho-kinase-dependent growth-discouraging activity is increased even in minor neurites in a neurite length-dependent manner. We have carefully described the results on page 13, lines 11-14 in the revised manuscript.

According to the reviewer's suggestion, we performed quantification analysis of each time point during photoactivation. The photoactivation of LOVTRAP-RhoA or LOVTRAP-Rho-kinase in the cell body significantly induced rapid minor neurite

retraction but not axon retraction from the first time point (approximately 10 min) compared to LOVTRAP-Cont and LOVTRAP-Rho-kinase KD. We have described the results on pages 10-11, changed the image of the photoactivated neurons in Fig. 3f and have added these results to Fig. 3g and Fig. 3h in the revised manuscript.

We thank the reviewer for this important comment. The sustained activation of RhoA in minor neurites from approximately 3–5 min to the end of the observation period occurred prior to minor neurite retraction (Supplemental Fig. 2 and Fig. 1). However, considering that CaMKI was activated by NT-3-induced long-range Ca^{2+} propagations in the cell body but not in minor neurites (Fig. 2e) and that the phosphorylation of GEF-H1 was not changed by NT-3 in the minor neurites (Supplemental Fig. 6f), RhoA was activated by the long-range inhibitory signaling in the cell body, and therefore, the activated RhoA might diffuse to minor neurites. We have discussed these issues in the revised manuscript (Pages 24-25).

4) To test diffusion of activated Rho-kinase into short minor neurites vs. the axon, the authors develop a mathematical model to show that Rho-kinase accumulates to a greater degree in minor neurites than in axons. This is shown by computer simulations, which I am unable to evaluate. The model for neurite retraction suggests that Rho-kinase regulated growth force is length-dependent so Rho-kinase would affect short but not long neurites, consistent with its ability to diffuse into short minor neurites but not axons.

We appreciate your efforts toward understanding our work.

5) In Figure 5 the authors ask whether prevention of outgrowth of minor neurites by active Rho-kinase is the mechanism for ensuring formation of a single long axon. Using gradients of a Rho-kinase inhibitor, the authors show with live cell imaging that this promotes rapid growth of a minor neurite. The video nicely shows this and reveals that the typical extension and retraction behavior does not occur. Can the authors comment on whether this steady forward extension is typical of minor neurites treated with the Rho-kinase inhibitor?

This figure shows that application of Rho-kinase inhibitors to hippocampal neurons increases the numbers of neurons with multiple axons. Interestingly as shown in supplemental Fig 4, newly formed tau labeled axons grow from the tips of minor neurites identified with MAP2. Since the two markers overlap it raises the question of how the new process is converted from a minor neurite to an axon.

As the reviewer pointed out, we have performed local application experiments based on a generally accepted protocol (Nakamuta et al., 2011). We have also confirmed the effectively decreased phosphorylation of MYPT-1 by Rho-kinase in neurons, indicating the inhibitor has been working (Figure for editor and referees). Thus, local application of Rho-kinase inhibitor (Y-27632) typically induced minor neurite elongation.

We agree with the reviewer's second comment regarding the important question of how the axon is generated from a minor neurite. It has been reported that pharmacological stabilization of microtubules or depolymerization of the actin filaments converts minor neurites into growing axons (Bradke and Dotti., 2000; Tahirovic and Bradke., 2009; Witte et al., 2008), indicating that the rearrangements of microtubules and the actin cytoskeleton are essential for the generation of an axon. Rho-kinase plays a critical role in the dynamics of actin filaments and microtubules through several substrates including LIM kinase, CRMP-2 and Tau (Amano et al., 2010; Tahirovic and Bradke., 2009; Govek et al., 2005). Thus, active Rho-kinase appears to regulate neuronal polarity through multiple pathways that modulate microtubule stabilization and actin filament dynamics. This interesting issue has been described on pages 24-25 in the Discussion of the revised manuscript.

Figure for editor and referees

Figure for editor and referees

Hippocampal neurons at 3 DIV were treated with Y-27632 (10 μ M and 20 μ M) for 48 h. Cell lysates were analyzed by immunoblotting with anti-pMYPT-1, which can detect the Rho-kinase phosphorylation, and anti-MYPT-1 antibodies.

6) In Fig 6 the authors investigate interaction between CAMKI and RhoA/Rho-kinase. Using a novel screening technique developed by their laboratory, the authors identify a number of candidate phospho proteins as substrates for CAMKI but focus on GEF-H I as the only specific RhoA regulator. The authors identify the Thr103 site on GEF-HI directly phosphorylated by CaMKI. Under physiological conditions when hippocampal neurons are stimulated with NT-3 GEF-H1 phosphorylation at Thr103 is increased and this increase in the cell body is clearly demonstrated in Supplemental figure 6.

We appreciate your efforts toward understanding our work.

7) As shown in Fig 7 the GEF-HI is required for the activation of RhoA, which underlies the retraction of minor neurites, and the computer model shows diffusion of activated Rho-kinase in minor neurites so I wonder whether the large increase in GEF-HI phosphorylation in the cell body also occurs in the minor neurites. Do the authors ever observe this? GEF-HI knockdown (supplemental Fig 7) or transfection of hippocampal neurons with a non phosphorylatable mutation of GEF-HI at T103 (Fig 7g) caused formation of multiple axons. Would the elimination of the restraint on minor neurites and the resulting growth of multiple axons by GEF-HI inactivation require its expression in minor neurites?

As the reviewer pointed out, we have carefully reexamined and performed the quantification analysis. The cell bodies of the NT-3-stimulated neurons showed slightly but significantly elevated amounts of phosphorylated GEF-H1, whereas axons and minor neurites did not. We have described the results on page 17, lines 7-10 in the revised manuscript, and added these results to Supplemental Fig. 6d-f.

As the reviewer pointed out, we have shown that the phosphorylation of GEF-H1 by CaMKI enhanced its GEF activity and suppressed axon formation *in vitro* and *in vivo* (Fig. 7 and Fig. 8). On the other hand, the non-phosphorylation mutant of GEF-H1 induced the formation of multiple axons (Fig. 8), which was similar to knockdown of GEF-H1, indicating that GEF-H1 maintains a single axon formation by preventing the formation of multiple axons depending on its phosphorylation state. However, we cannot exclude the possibility that the non-phosphorylation mutant of GEF-H1 would have functioned as a dominant-negative factor. Thus, GEF-H1 plays a critical role in the formation of a single axon not only through its expression but also through its phosphorylation by CaMKI. We have discussed these issues in the revised manuscript (Page 23, lines 4-16).

8) In the final figure 8 (Supplemental) the authors test the relevance of this RhoA signaling pathway to the development of neuronal polarization and migration of cortical pyramidal neurons *in vivo*. Using *in utero* electroporation of GEF-H1 to knock down or express WT, phosphomimetic (T103E) and non-phospho (T103A) constructs, the authors found that overexpression of WT GEF-H1 or the T103 and T103A mutants all caused failure of migration into the cortical plate with most neurons lacking a leading process in contrast to control bipolar neurons. However, the morphology of the T103E vs T103A expressing neurons had very different morphologies i.e. round shapes with few processes vs. long trailing processes. How do the authors interpret these results in light of their *in vitro* findings? GEF-H1 knockdown also reduced pyramidal neuron migration into the cortical plate and fewer neurons had bipolar morphologies. These defects were rescued with

GEF-H1 WT electroporation. In my view these *in vivo* experiments are important and should be presented in the body of the paper with a fuller discussion rather than as a supplemental figure.

We have clearly shown that the expression of GEF-H1-T103E inhibited axon formation, whereas the expression of GEF-H1-T103A induced Tau-1-positive multiple axon formation and, thereby, prevented dendritic specification in cultured hippocampal neurons (Fig. 7). Consistently, GEF-H1-T103E, but not GEF-H1-T103A, suppressed trailing process (future axon) formation in the developing cortex. On the other hand, GEF-H1-T103A-expressing pyramidal cells formed unidentified multiple processes, which were shorter than that of trailing processes, resulting in an interruption of leading process (future dendrite) formation *in vivo*. These results indicate that phosphorylation of GEF-H1 by CaMKI plays a critical role in axon-dendrite polarization *in vitro* and *in vivo*. This issue has been fully discussed on page 23, lines 4–5, pages 25-26, and we have added these results to Fig. 8 in the revised manuscript.

9) This interesting paper uses a variety of techniques to present a novel and mechanistic model of how neurotrophins, by activating a retrograde inhibitory calcium signaling pathway involving RhoA contractility of minor neurites, can induce polarity in hippocampal neurons resulting in a single long axon and multiple short dendrites. Given the extensive and convincing data in the paper, I found the Discussion somewhat flat in that the authors mostly summarized the results. For example the authors raise an important question regarding development of neuronal polarity in the absence of exogenous neurotrophins (i.e. a stochastic model) but do not fully address this issue. Would similar signaling pathways occur in the absence of neurotrophins? It would also be interesting for the authors to discuss how their intriguing *in vivo* results on cortical migration and neuronal morphology relate to their findings in cultured hippocampal neurons, which was not clear from their

description of supplemental figure 8. Nevertheless this paper should be of wide interest to researchers interested in the important mechanisms of neuronal polarity.

We agree with the reviewer's important comment regarding the effect of Ca^{2+} -mediated long-range inhibitory signaling on neuronal polarity in the stochastic model, which is related to Reviewer #1, comment 2. By local application of the specific inhibitors, we found that the attenuation of long-range Ca^{2+} propagations from the axon to the cell body and the following CaMKI activity at the cell body induced minor neurite elongation (Fig. 2 and Supplemental Fig. 1). Moreover, local inhibition of Rho/Rho-kinase in minor neurites induced rapid neurite elongation and subsequent multiple axon formation (Fig. 5). Therefore, these results indicate that Ca^{2+} /CaMKI/GEF-H1/RhoA/Rho-kinase signaling plays a critical role in neuronal polarization not only in the NT-3-stimulated model but also in the stochastic model. This issue has been discussed on page 22, lines 12–18 in the revised manuscript.

According to the reviewer's suggestion, we have added the discussion regarding the effect of long-range inhibitory signaling on neuronal polarization *in vivo*. We previously reported that CaMKI and RhoA/Rho-kinase regulates MP-to-BP transition in the developing neocortex and proper neuronal entry into CP (Nakamuta et al., 2011; Xu et al., 2015). However, the regulatory interaction between CaMKI and RhoA/Rho-kinase during neuronal polarization *in vivo* remains undetermined. Thus, here, we focused on GEF-H1 functions and found that GEF-H1 and its phosphorylation were required for the MP-to-BP transition and the subsequent neuronal migration (Fig. 8). The expression of GEF-H1-T103E impaired MP-to-BP transition *in vivo*, which was consistent with previous results of active CaMKI and RhoA (Nakamuta et al., 2011; Pacary et al., 2011). We recently found that expression of the dominant-negative form of RhoA or Rho-kinase also impairs MP-to-BP transition and neuronal migration (Xu et al., 2015). The appropriate activity of RhoA/Rho-kinase is vital for neuronal polarization and, therefore, might be properly regulated by CaMKI-mediated phosphorylation of GEF-H1 in the developing

neocortex. Thus, CaMKI/GEF-H1/RhoA/Rho-kinase signaling is a novel signaling pathway that function in long-range inhibition to establish and maintain neuronal polarity during neuronal development. We have included this discussion in the revised manuscript (Pages 25-26).

Reviewer #3:

Thank you for your valuable comments. We have revised the manuscript accordingly.

1) In Figure 1, the authors locally applied two neurotrophins gradient, NT-3 and BDNF, to enhance the inhibitory signals from axon. However, no attempt was made to test the in vivo relevance about the neurotrophin gradient of these neurotrophins. Do these factors distributed in a gradient in hippocampus and developing cortex in vivo?

To our knowledge, the gradient expression pattern of neurotrophins remains unknown. Neurotrophins are highly basic molecules that tend to be associated with the cell surface after secretion, thereby serving as autocrine factors (Nakamuta et al., 2011; Cheng et al., 2011; Cheng and Poo., 2012). Neurotrophins secreted by one neuron also act on nearby neurons in a paracrine fashion *in vivo* (Takano et al., 2015; Namba et al., 2015). We have described this in the introduction on page 3, lines 14-17 in the revised manuscript.

2) In Figure 2, calcium wave was observed after local application of NT-3 to single axon. The treatment of four different inhibitors suggests that ryanodine receptor is involved in Ca-wave transition through axon to cell body/dendrite. However, the experiments that test how these inhibitors affect axon and minor neurites outgrowth are missing.

According to the reviewer's suggestion, we examined whether the long-range Ca²⁺ waves generated from ryanodine receptors were implicated in NT-3-induced minor

neurite retraction. All of the inhibitors, including ryanodine and dantrolene (ryanodine receptors inhibitors), abolished the minor neurite retraction induced by NT-3. On the other hand, treatment with xestospongins C (IP₃ receptor inhibitor) but not ryanodine, dantrolene (ryanodine receptor inhibitors) or SKF96365 (TRP channel inhibitor) canceled the NT-3-induced axonal elongation, which is consistent with previous results (Nakamuta et al., 2011). These results indicate that long-range Ca²⁺ waves are required for NT-3-induced minor neurite retraction. We have described these results on pages 6-7 and have added these results to Fig. 2c-d in the revised manuscript.

3) In Figure 3, the authors used Raichu-RhoA probe to monitor RhoA signaling activity. The authors mentioned “RhoA activity” in the main text (page 8, line 136), however, this probe can detect the activity of GEF against RhoA (more precisely, the balance of activity between GEF and GAP), not directly detect RhoA activity. This needs to be clearly described.

According to the reviewer’s suggestion, we have provided more understandable information about Raichu-RhoA-CR in the revised manuscript (Pages 8-9).

4) In Figure 4, the authors used computational simulation to monitor ROCK distribution after photoactivation (in Figure 3). Simulation appears to be an excellent way to monitor ROCK distribution, but in this case, ROCK-Zdk is overexpressed in the neurons, which unlikely represent endogenous ROCK localization. It would be better to simulate endogenous ROCK localization under control (stochastic model) and NT-3-treated condition, if possible. Is there any good antibody to detect endogenous ROCK? Moreover, the authors should demonstrate the level of ROCK overexpression in comparison to endogenous ROCK.

According to the reviewer's suggestion, we investigated the spatial distribution of endogenous Rho-kinase in locally stimulated neurons with anti-Rho-kinase (Iizuka et al., 2012). The 2.5-D reconstruction analysis clearly showed that Rho-kinase was found in stage 3 hippocampal neurons to be localized throughout the cytoplasm but to be prominent in minor neurites and in the proximal region of the axon. Interestingly, when NT-3 was locally applied to the axon, Rho-kinase was enriched in the cell body and minor neurites compared with the axon, which is similar to the results of the Rho-kinase diffusion model (Fig. 4c). We have described the results on page 14, lines 4-11, and have added these results to Supplemental Fig. 5a in the revised manuscript.

5) In Figure 7, the authors showed that WT GEF-H1 can be localized in not only cell body/dendrite, but also axon shaft. This raises another questions why RhoA-ROCK-shrinkage signaling is not activated in an axon, because Ca and CaMKI are also activated in axons after local application of NT-3 to axons. This needs to be addressed.

We thank the reviewer for this important comment. The mechanism by which GEF-H1 regulates polarized RhoA activity in neurons remains unknown. One possibility is that the polarized activity of RhoA results from the difference in phosphorylation states of GEF-H1 in neurons. Indeed, the phosphorylation of GEF-H1 at Thr103 was increased by NT-3 in the cell body but not in the axon or minor neurites (Supplemental Fig. 6c-f). Another possibility is that GEF-H1 may be specifically inactivated in the axon by PKA because PKA has been activated in the growing axon by neurotrophins (Meiri et al., 2009; Shelly et al., 2007). This interesting issue has been described on page 24, lines 2-8, in the Discussion of the revised manuscript.

6) To assess the GEF-H1 effect on axon-dendrite polarity in vivo, the authors performed in utero electroporation into telencephalon. This is an excellent experiment, however, the authors should also evaluate CaMKI, RhoA and ROCK function in these experimentsn to test the role of this signaling axis.

We thank the reviewer for this important comment, which is related to comment 9 from Reviewer #2. We previously reported that CaMKI and RhoA/Rho-kinase regulate the MP-to-BP transition in the developing neocortex and proper neuronal entry into CP (Nakamuta et al., 2011; Xu et al., 2015). However, the regulatory interaction between CaMKI and RhoA/Rho-kinase during neuronal polarization *in vivo* remains undetermined. Thus, here we focused on GEF-H1 functions and found that GEF-H1 and its phosphorylation were required for the MP-to-BP transition and the subsequent neuronal migration (Fig. 8). The expression of GEF-H1-T103E impaired MP-to-BP transition *in vivo*, which is consistent with previous results of active CaMKI and RhoA (Nakamuta et al., 2011; Pacary et al., 2011). We recently found that expression of the dominant-negative form of RhoA or Rho-kinase also impairs MP-to-BP transition and neuronal migration (Xu et al., 2015). The appropriate activity of RhoA/Rho-kinase is vitally essential for neuronal polarization, and therefore might be properly regulated by CaMKI-mediated phosphorylation of GEF-H1 in developing cortex. We have discussed these important issues in the revised manuscript (Pages 25-26).

Reviewer #4:

We would like to appreciate the reviewer's invaluable and constructive comments. We have improved our manuscript and prepared a response to each of the reviewer's comments and suggestions, as given below.

Specific comments:

1. In Eq. (1), should it be $(\cosh(L/\sqrt{D/k}))^{-1}$ (which is $\operatorname{sech}(L/\sqrt{D/k})$) instead of $\cosh^{-1}(L/\sqrt{D/k})$ (which is $\operatorname{arcosh}(L/\sqrt{D/k})$)?

As pointed out by the reviewer, the expression of equation (1) was inappropriate. In the revised manuscript, we re-described this equation as $R(L) = (\cosh(L/\sqrt{(D/k)}))^{-1}$. This was simply a problem of expression, and thus, this revision does not affect our result.

2. The authors assume that the reaction-diffusion process attains steady state rapidly compared to the time scale of optogenetic experiments (~60 minutes). Based on $\sqrt{(D/k)} = 31.6 \mu\text{m}$ (given in figure 4 legend), the half-life of active Rho-kinase $(\ln(2)/k)$ works out to be ~11.5 minutes (for $D = 1 \mu\text{m}^2 \text{s}^{-1}$) and ~115 minutes (for $D = 0.1 \mu\text{m}^2 \text{s}^{-1}$). The half-life appears to be comparable to the time scale of experiments. Thus, it is not clear if the system attains steady state rapidly. The authors should describe the parameter regime (D and k) in which their steady state assumption holds and discuss if the parameter regime is biologically reasonable.

The reviewer is concerned regarding our assumption that the distribution of Rho-kinase along the neurite almost reached steady state in equation (1). In a previous report, we only mentioned $\sqrt{(D/k)} = 31.6 (\mu\text{m})$ but did not specify individual values of D and k . In the revised manuscript, we adopted $D = 20 (\mu\text{m}^2/\text{s})$ and $k = 0.02 (\text{s}^{-1})$, i.e., $\sqrt{(D/k)} = 31.6 (\mu\text{m})$. These parameter values are within reasonable scales. The diffusion rate of protein in mammalian cell cytoplasm is 20~100 ($\mu\text{m}^2/\text{s}$) (Kühn et al., 2011). In response to photoactivation, Rho-kinase is rapidly released in less than one minute (Wang et al., 2016), which means that its time constant is approximately 60 seconds, (i.e., $k = 1/60 \approx 0.02 (\text{s}^{-1})$). In the revised manuscript, we mentioned these points (see pages 33, lines 5-10). By simulation using these parameter values, we also confirmed that the Rho-kinase concentration at the neurite tip rapidly became stationary (~100 seconds) (Supplemental Fig. 4a) (see pages 12, lines 15-18).

In addition, the reviewer suggested us to describe the parameter regime of D and k that satisfies our steady state assumption. To this end, we numerically investigated the time required for the Rho-kinase concentration at the neurite tip to reach half the

stationary value from the onset of photoactivation (Supplemental Fig. 4b). We then showed that our steady state assumption is reasonable unless both D and k are too small (Supplemental Fig. 4c).

Furthermore, we performed a reaction-diffusion simulation along a growing neurite without the steady state approximation. This analysis clearly showed that the simulated results with and without the approximation were almost the same (Supplemental Fig. 4d-f). Therefore, our steady state assumption (approximation) was validated.

Note that computational cost of the reaction-diffusion simulation was high, and therefore, it is not suitable for parameter estimation, which requires many iterations of simulation (Fig. 4h). Thus, we adopted the approximation.

3. Previous mathematical models of axon-dendrite specification consider both diffusion and active transport (anterograde and retrograde) of molecules between the soma and the neurite tips (for e.g., refs. [1-3]). Active transport could enrich molecules at the axon tips. Is there a reason to neglect transport of active RhoA kinase?

The reviewer asked why our model does not include active transport of Rho-kinase from the soma to the neurite tips, in contrast to previous models. According to our previous theoretical work (Naoki et al., 2008), actively transported molecules are more accumulated in longer neurites. If this was the case for Rho-kinase, the growth of longer neurites would be more inhibited by active Rho-kinase. However, this is inconsistent with our experimental finding that the growth of shorter neurites was more inhibited by photoactivation of Rho-kinase. Thus, we did not consider active transport of Rho-kinase. Moreover, in our imaging experiments, we did not observe active transport of Rho-kinase.

4. The neurite growth model (Eq. (2)) predicts that, during photoactivation of Rho-kinase, the neurite length does not change for large values of initial neurite length (Figs. 4e and 4h).

This prediction appears to be inconsistent with the experimental data where the axon length appears to increase during photoactivation (Fig. 3g). Shouldn't axons grow at the normal rate when the photoactivated Rho-kinase level is very low at the axon tips?

The reviewer indicated that our simulation did not reproduce experimentally-observed elongation of long neurites during the photoactivation of Rho-kinase. This result was due to the initial condition we set, for which neurite lengths already reached steady state before the photoactivation, i.e., $L_i(t=0)=(F_o+F_i)/\mu$. In the revision, by modifying the initial condition such that neurite lengths almost reached steady state but continued growing (see page 33, line 5), we successfully reproduced the experimental observation (Fig. 4h).

5. The rate $-\mu L_i$ in the Eq. (2) indicates that the neurite retraction rate increases with neurite length. I find it hard to understand why retraction rate should increase with neurite length. Is there any direct experimental evidence or biophysical reasoning for this assumption?

Yes, there is one piece of experimental evidence. Lamoureux et al., Nature, 1989, clearly showed that neurite tension linearly increases with neurite length (growth cone advance). The neurite tension contributes to neurite retraction. Therefore, the neurite retraction rate proportional to neurite length should be valid. In the revision, we mentioned this concept by citing this reference (see page 12, lines 17-18). In addition, we biophysically speculated that neurite tension increases with neurite length because the amount of membrane is conserved (or limited) at least at the initial phase of neuronal polarization (see page 12, lines 17-18).

6. The parameter μ in the Eq. (2) is defined as Young's modulus for retraction force. The unit of Young's modulus is the pascal, whereas the unit of μ in Eq. (2) is s⁻¹. It is unclear what μ means in the equation.

We apologize for the inadequate usage of Young's modulus. We revised the notation of μ by "retraction coefficient" (see page 12, lines 11-12). In the revision, we present the derivation of Eq. (2), in which μ is explained in relation to the spring constant (see pages 33-34), as Young's modulus corresponds to spring constant divided by the cross-section of the material of interest, e.g., neurite.

7. As per the Eq. (2), the maximum length that any neurite can grow is $(F_0 + F_i)/\mu$ (assuming $R_i = 0$). It is not clear if such a limit exists for neurite growth, as axons can grow until synaptic connections are established.

As indicated by the reviewer, the model neurites have an upper limit of length due to the linear relationship between neurite retraction rate and neurite length. We understood that this is unlikely when neurons extend very long axons (e.g., 10cm~1m) to make synaptic connections. However, according to Lamoureux et al., Nature, 1989, such a linear relationship is held within ~140 μm (initial length=57 μm ; maximum growth cone advance=83 μm). We expected that very long neurites loose tension to stably maintain their length. Thus, our model only works at the initial phase of neuronal development.

8. The authors estimate six model parameters (μ , F_0 , F_i , $\sqrt{(D/k)}$, h and K/C_0) from one data set (Fig. 4h). The authors must demonstrate that they are not overfitting the data. Perhaps the authors could use the control experiments (Fig. 3f) to constrain some of the model parameters such as μ and F_0 .

The reviewer is concerned regarding an overfitting problem. We know that if the fitting function is a high-order polynomial function or an artificial neural network, these functions can overfit the data, losing prediction ability for new data. However, the fitting function in our case was constructed by the model simulation and thus restricted by the model structure such that the function does not have the potential to

overfit the data. In fact, the black line in Fig. 4h does not perfectly overlap all data points.

Furthermore, in the revision, we checked whether overfitting occurred in our model. We divided the data points into training and test data sets, which were used to estimate parameters and evaluate overfitting, respectively. We then showed high correlation between the observed and predicted test data sets, which indicated no overfitting issue (Figure for editor and referees).

Figure for editor and referees

Figure for editor and referees

Relationship between observed and predicted neurite retractions. Neurite retractions in the test data set were predicted by the function with the estimated parameters, i.e., black line in figure 4h. Observed and predicted neurite retractions were highly correlated, indicating no overfitting problem in the parameter estimation.

9. The authors use $P = 0.1$ as a threshold for statistical significance in Fig. 1 and Supplementary Fig. 1, and $P = 0.05$ elsewhere. The authors will have to justify this in the manuscript.

We have carefully checked the P values of all statistical data and found that some of them were incorrect. We have corrected the P values in Fig. 1 and Supplemental Fig. 1 and added a table providing the summary of the statistical analysis to Supplemental Table 2 in the revised manuscript.

10. How did the authors determine the sample size of their experiments? The authors must discuss in the manuscript whether the chosen sample size is sufficient to measure their effect size (see ref. [4]).

The sample size was defined based on our previous reports (Nakamuta et al., 2011; Namba et al., 2014) and G*Power 3.1 software. According to the reviewer's suggestion, the effect size of all statistical data was calculated (Sullivan and Feinn, 2012). We found that the effect size was shown to be more than the medium effect (Cohen's $d > 0.5$), and their sample sizes were sufficient. We have provided more understandable information about the statistical analysis on pages 38-39, and we have added these results to Supplemental Table 2 in the revised manuscript.

11. It is not clear if the n values in the figure legends indicate number of cells or number of independent experiments. Please mention both in the figure legends.

According to the reviewer's suggestion, we have clearly described the n values of all statistical data in Supplemental Table 2 in the revised manuscript.

Minor points:

1. Page 44, line 789, equation number should be (5) instead of (4).

We corrected this typo. Thank you for pointing it out.

2. Please provide all the estimated parameter values (for instance, F_i value is missing).

In Fig. 4h, we estimated the values of K , h and F_o , which were already provided. F_i was not a parameter to be estimated but rather a control parameter such that it determines neurite length. In Fig. 4f and h, the black lines were plotted for increasing F_i values. The other parameter values ($\sqrt{D/k}$, τ) were the same as in Fig. 4b, c, e, f, as those values are biologically reasonable. In the revision, we clearly stated these points (see Figure legend of Fig. 4).

References

- Nakamuta, S. et al. Local application of neurotrophins specifies axons through inositol 1,4,5-trisphosphate, calcium, and Ca²⁺/calmodulin-dependent protein kinases. *Sci. Signal.* 4, ra76 (2011).
- Cheng, P.L. et al. Self-amplifying autocrine actions of BDNF in axon development. *Proc. Natl. Acad. Sci. USA* 108, 18430-5 (2011).
- Arimura, N. et al. Anterograde transport of TrkB in axons is mediated by direct interaction with Slp1 and Rab27. *Dev. Cell* 16, 675-86 (2009).
- Randriamampita, C. et al. Ca²⁺-induced Ca²⁺ release amplifies the Ca²⁺ response elicited by inositol trisphosphate in macrophages. *Cell Regul* 2, 513-522 (1991).
- Kochegarow, AA. et al. Ionomycin and 2,5'-di(tertbutyl)-1,4,-benzohydroquinone elicit Ca²⁺-induced Ca²⁺ release from intracellular pools in *Physarum polycephalum*. *Comp. Biochem Physiol A Mol Integr Physiol* 128, 279-288 (2001).
- Wang, H. et al. LOVTRAP, An Optogenetic System for Photo-induced Protein Dissociation. *Nat. Methods.* (2016).
- Collier, ML. et al. Calcium-induced calcium release in smooth muscle: loose coupling between the action potential and calcium release. *J Gen Physiol* 115, 653-662 (2000).
- Nakamura, T. et al. Spatial segregation and interaction of calcium signalling mechanisms in rat hippocampal CA1 pyramidal neurons. *J Physiol* 543, 465-480 (2002).

Ross, MN. Understanding calcium waves and sparks in central neurons. *Nat Rev Neurosci.* 13, 157-168 (2012).

Neymotin, SA et al. Neuronal calcium wave propagation varies with changes in endoplasmic reticulum parameters: a computer model. *Neural Comput* 27, 898-924 (2015)

Mori, K. et al. Rho-kinase contributes to sustained RhoA activation through phosphorylation of p190A RhoGAP. *J Biol Chem* 284, 5067-5076 (2009).

Takano, T., Xu, C., Funahashi, Y., Namba, T. & Kaibuchi, K. Neuronal polarization. *Development* 142, 2088-93 (2015).

Amano, M. et al. Kinase-interacting substrate screening is a novel method to identify kinase substrates. *J. Cell Biol.* 209, 895-912 (2015).

Bradke, F. & Dotti, C.G. Differentiated neurons retain the capacity to generate axons from dendrites. *Curr. Biol.* 10, 1467-70 (2000).

Tahirovic, S. & Bradke, F. Neuronal polarity. *Cold Spring Harb Perspect Bio.* 1, a001644 (2009)

Witte, H. & Bradke, F. The role of the cytoskeleton during neuronal polarization. *Curr. Opin. Neurobiol.* 18, 479-87 (2008).

Amano, M., Nakayama, M. & Kaibuchi, K. Rho-kinase/ROCK: A key regulator of the cytoskeleton and cell polarity. *Cytoskeleton (Hoboken)* 67, 545-54 (2010).

Tahirovic, S. & Bradke, F. Neuronal polarity. *Cold Spring Harb Perspect Bio.* 1, a001644 (2009).

Govek, E.E., Newey, S.E. & Van Aelst, L. The role of the Rho GTPases in neuronal development. *Genes Dev.* 19, 1-49 (2005).

Xu, C. et al. Radial Glial Cell-Neuron Interaction Directs Axon Formation at the Opposite Side of the Neuron from the Contact Site. *J. Neurosci.* 35, 14517-32 (2015).

Pacary, E. et al. Proneural transcription factors regulate different steps of cortical neuron migration through Rnd-mediated inhibition of RhoA signaling. *Neuron.* 24, 1069-1084 (2011).

Cheng, P.L. & Poo, M.M. Early events in axon/dendrite polarization. *Annu. Rev. Neurosci.* 35, 181-201 (2012).

Namba, T. et al. Extracellular and Intracellular Signaling for Neuronal Polarity. *Physiol. Rev.* 95, 995-1024 (2015).

Iizuka, M. et al. Distinct distribution and localization of Rho-kinase in mouse epithelial, muscle and neural tissues. *Cell Struct Funct* 37, 155-175 (2012).

Meiri, M. et al. Modulation of Rho guanine exchange factor Lfc activity by protein kinase A-mediated phosphorylation. *Mol Cell Biol* 29, 5963-5973 (2009).

Shelly, M. et al. LKB1/STRAD promotes axon initiation during neuronal polarization. *Cell* 129, 565-577 (2007).

Kühn, T et al. Protein Diffusion in Mammalian Cell Cytoplasm. *PLoS ONE* 6, e22962 (2011).

Naoki, H., Nakamuta, S., Kaibuchi, K. & Ishii, S. Flexible search for single-axon morphology during neuronal spontaneous polarization. *PLoS One* 6, e19034 (2011).

Lamoureux, P et al. Direct evidence that growth cones pull. *Nature* 340, 159-162 (1989).

Namba, T. et al. Pioneering axons regulate neuronal polarization in the developing cerebral cortex. *Neuron* 81, 814-29 (2014).

Sullivan, GM. & Feinn, M. Using Effect Size-or Why the P Value Is Not Enough. *J Grad Med Educ* 4, 279-282 (2012).

Reviewers' comments:

Reviewer #1 (Remarks to the Author):

The revised manuscript addressed most of the comments with either text editing or new experiments. Most revisions are satisfactory but there are still some concerns with the original comment 1 and 2. To my opinion, in vitro model of neuronal polarization typically means the formation of a single axon among multiple neurites. After that, the formed axon continues polarized growth. The reported study specifically investigated the polarized axon growth step rather than the first polarization step (initial axon formation step). The control neurons without NT3 application obviously polarize normally and grow axons normally. Does local NT3-calcium signaling (through autocrine) also underlie polarized axon growth (not initial axon formation step) in control neurons? The answer to this question is important to confirm that the identified pathway and mechanism are physiological relevant and important.

Reviewer #2 (Remarks to the Author):

In this revised manuscript the authors address the important question of how polarity is established in hippocampal neurons. The authors show that neurotrophins activate a retrograde calcium signaling pathway that inhibits growth of minor neurites resulting in a single long axon. I have carefully read the authors responses to all of the reviewers' comments and questions, which the authors have carefully and thoroughly considered. The manuscript is greatly improved by revisions in Results, Discussion and Figures. I have no further concerns and consider the novel results of the manuscript to be convincingly documented and of wide interest to the field.

Reviewer #3 (Remarks to the Author):

The authors have satisfied my critiques and this manuscript is ready for publication.

Reviewer #4 (Remarks to the Author):

The authors have addressed my concerns to a large extent, and I think the model provides a reasonable interpretation of the LOVTRAP experiments. I have the following comments and suggestions for the authors to consider.

1. As mentioned in my previous comment #8, perhaps comparing the model predictions with the LOVTRAP control experiments would be a valuable addition/validation.
2. Please clarify why the estimated parameter values are different in the legend of figure 4 and in the methods section (page 33).

3. In the revised version, the authors have clarified that F_i is a control parameter (and not a fit parameter) used to vary the initial neurite length in their simulations. I think this implies that longer the neurite larger is the growth force. If this is true, please clarify why this assumption is valid in the manuscript.

4. Their explanation to ignore active transport of RhoA kinase in the model seems reasonable. It would be helpful if the authors could comment on this in the manuscript, as previous models of axon-dendrite specification consider active transport.

5. The authors now provide the number of neurites analysed in each condition in the supplementary table 2. It would help if the authors could also mention the number of independent experiments performed in each condition.

Our responses to the editors' suggestion and the reviewer's comments in the revised version

Thank you very much for reviewing our manuscript. We appreciate your efforts toward understanding our work and your valuable comments on the manuscript. We have attempted to address all the questions and concerns.

Reviewer #1 (Remarks to the Author):

We greatly appreciate your positive comments on our revised manuscript.

The revised manuscript addressed most of the comments with either text editing or new experiments. Most revisions are satisfactory but there are still some concerns with the original comment 1 and 2. To my opinion, in vitro model of neuronal polarization typically means the formation of a single axon among multiple neurites. After that, the formed axon continues polarized growth. The reported study specifically investigated the polarized axon growth step rather than the first polarization step (initial axon formation step). The control neurons without NT3 application obviously polarize normally and grow axons normally. Does local NT3-calcium signaling (through autocrine) also underlie polarized axon growth (not initial axon formation step) in control neurons? The answer to this question is important to confirm that the identified pathway and mechanism are physiological relevant and important.

According to the reviewer's suggestion, we examined whether the local amplification of NT-3 derived from cultured neurons in a nascent axon enhances axon formation in polarized neurons. We found that local application of a neutralizing antibody against NT-3 to the axon terminal inhibited axon outgrowth. In the previous revised manuscript, we have shown that local application of a Ca²⁺ signaling inhibitor (xestopongin C), which inhibits the elevation of Ca²⁺

concentration through IP₃ receptors, inhibited axon elongation. These results suggest that the local NT-3-Ca²⁺ signaling is required for axon formation in the stochastic model. We have described these results (page 5, lines 2-7, page 8, lines 9-11, and Supplementary Fig. 1) in the revised manuscript.

Reviewer #2 (Remarks to the Author):

In this revised manuscript the authors address the important question of how polarity is established in hippocampal neurons. The authors show that neurotrophins activate a retrograde calcium signaling pathway that inhibits growth of minor neurites resulting in a single long axon. I have carefully read the authors responses to all of the reviewers' comments and questions, which the authors have carefully and thoroughly considered. The manuscript is greatly improved by revisions in Results, Discussion and Figures. I have no further concerns and consider the novel results of the manuscript to be convincingly documented and of wide interest to the field.

We greatly appreciate your positive comments on our revised manuscript.

Reviewer #3 (Remarks to the Author):

The authors have satisfied my critiques and this manuscript is ready for publication.

We greatly appreciate your positive comments on our revised manuscript.

Reviewer #4 (Remarks to the Author):

The authors have addressed my concerns to a large extent, and I think the model provides a reasonable interpretation of the LOVTRAP experiments. I have the following comments and suggestions for the authors to consider.

We would like to thank for positively evaluating our previous revision. We have improved our manuscript and prepared the response to each of the reviewer's comments and suggestions as shown below.

1. As mentioned in my previous comment #8, perhaps comparing the model predictions with the LOVTRAP control experiments would be a valuable addition/validation.

According to the reviewer's comment, we simulated the LOVTRAP control condition using the estimated parameters (Fig. 4h; note we integrated previous Fig. 4g and h into new Fig. 4g). We then confirmed that the model reproduced a positive correlation between neurite length at the onset of photoactivation and neurite growth length after photoactivation (page 14, lines 5-8).

2. Please clarify why the estimated parameter values are different in the legend of figure 4 and in the methods section (page 33).

We are sorry for confusing you. Parameter values listed in the methods section were correct. We deleted those listed in the legend of figure 4.

3. In the revised version, the authors have clarified that F_i is a control parameter (and not a fit parameter) used to vary the initial neurite length in their simulations. I think this implies that longer the neurite larger is the growth force. If this is true, please clarify why this assumption is valid in the manuscript.

This is the case. We assumed that the growth force (F_i) was regulated by a set of molecules, e.g., Shootin1 and H-Ras (Toriyama et al., 2010; Fivaz et al., 2008), accumulated at the neurite tip through active transport. In fact, these molecules exhibit greater accumulation at the tips of long neurites compared with short neurites (Toriyama et al., 2010; Fivaz et al., 2008). Consistently, in our model, the

longer neurite results from larger growth force. In the revised manuscript, we have provided the information (page 13, lines 7-11).

4. Their explanation to ignore active transport of RhoA kinase in the model seems reasonable. It would be helpful if the authors could comment on this in the manuscript, as previous models of axon-dendrite specification consider active transport.

As suggested by the reviewer, we explained validity to ignore active transport of Rho-kinase in the revised methods (page 33, lines 3-10).

5. The authors now provide the number of neurites analysed in each condition in the supplementary table 2. It would help if the authors could also mention the number of independent experiments performed in each condition.

According to the reviewer's suggestion, we have clearly described the number of independent experiments in Supplemental Table 2 in the revised manuscript.

References

Toriyama, M., Sakumura, Y., Shimada, T., Ishii, S. & Inagaki, N. A diffusion-based

length-sensing mechanism involved in neuronal symmetry breaking. *Mol. Syst.*

Biol **6**, 394 (2010).

Fivaz, M., Bandara, S., Inoue, T. & Meyer, T. Robust neuronal symmetry breaking by

Ras-triggered local positive feedback. *Curr. Biol* **18**, 44-50 (2008).

REVIEWERS' COMMENTS:

Reviewer #1 (Remarks to the Author):

I have no further concerns of the manuscript. It is now a very nice study ready for publication.

Reviewer #4 (Remarks to the Author):

The authors have addressed all my concerns regarding the modelling aspect of the paper, and I have no further comments.

Our responses to the editors' suggestion and the reviewer's comments in the revised version.

Thank you very much for reviewing our manuscript. We appreciate your efforts toward understanding our work and your valuable comments on the manuscript.

Reviewer #1:

I have no further concerns of the manuscript. It is now a very nice study ready for publication.

We greatly appreciate your positive comments on our revised manuscript.

Reviewer #4 (Remarks to the Author):

The authors have addressed all my concerns regarding the modelling aspect of the paper, and I have no further comments.

We greatly appreciate your positive comments on our revised manuscript.